# Novel sterol binding domains in bacteria

**Liting Zhai[1†], Amber C Bonds[2], Clyde A Smith[3], Hannah Oo[1], Jonathan Chiu-Chun Chou[1], Paula V Welander[2], Laura MK Dassama[1,4]***

[1]Department of Chemistry and Sarafan ChEM-H, Stanford University, Stanford, United States; [2]Department of Earth System Science, Stanford University, Stanford, United States; [3]Department of Chemistry and Stanford Synchrotron Radiation Lightsource, Stanford University, Stanford, United States; [4]Department of Microbiology and Immunology, Stanford University School of Medicine, Stanford, United States

**\*For correspondence:** dassama@stanford.edu

**Present address:** [†]School of Life Sciences, The Chinese University of Hong Kong, 14 Shatin, New Territories, Hong Kong, China

**Abstract** Sterol lipids are widely present in eukaryotes and play essential roles in signaling and modulating membrane fluidity. Although rare, some bacteria also produce sterols, but their function in bacteria is not known. Moreover, many more species, including pathogens and commensal microbes, acquire or modify sterols from eukaryotic hosts through poorly understood molecular mechanisms. The aerobic methanotroph *Methylococcus capsulatus* was the first bacterium shown to synthesize sterols, producing a mixture of C-4 methylated sterols that are distinct from those observed in eukaryotes. C-4 methylated sterols are synthesized in the cytosol and localized to the outer membrane, suggesting that a bacterial sterol transport machinery exists. Until now, the identity of such machinery remained a mystery. In this study, we identified three novel proteins that may be the first examples of transporters for bacterial sterol lipids. The proteins, which all belong to well-studied families of bacterial metabolite transporters, are predicted to reside in the inner membrane, periplasm, and outer membrane of *M. capsulatus,* and may work as a conduit to move modified sterols to the outer membrane. Quantitative analysis of ligand binding revealed their remarkable specificity for 4-methylsterols, and crystallographic structures coupled with docking and molecular dynamics simulations revealed the structural bases for substrate binding by two of the putative transporters. Their striking structural divergence from eukaryotic sterol transporters signals that they form a distinct sterol transport system within the bacterial domain. Finally, bioinformatics revealed the widespread presence of similar transporters in bacterial genomes, including in some pathogens that use host sterol lipids to construct their cell envelopes. The unique folds of these bacterial sterol binding proteins should now guide the discovery of other proteins that handle this essential metabolite.

## eLife assessment

This is a **valuable** contribution to our understanding of how some bacteria can transport sterols from the cytoplasm to the outer membrane. Though much remains to be tested and explored, the data and analyses presented here provide **solid** evidence for the genetic and physical interaction of BstA/B/C with bacterially-produced sterols. The manuscript will be of interest to scientists focusing on the characterization of novel bacterial proteins and those studying lipid transport and acquisition in bacterial pathogens.

## Introduction

Sterol lipids are ubiquitous and essential components of eukaryotic life, playing vital roles in intra- and intercellular signaling, stress tolerance, and maintaining cell membrane integrity (*Bi and Liao, 2010*;

*Bloch, 1991*; *Huang et al., 2016*; *Miao et al., 2002*). Although sterol synthesis is often considered to be a strictly eukaryotic feature, several bacteria have been shown to also produce sterols (*Bloch, 1991*; *Ourisson et al., 1987*). Bacterial sterols were first discovered in *Methylococcus capsulatus* more than 40 years ago (*Bird et al., 1971*) and initially, were only observed in a few isolated aerobic methanotrophs (γ-Proteobacteria) and a few myxobacteria (δ-Proteobacteria) (*Bode et al., 2003*; *Kohl et al., 1983*; *Patt and Hanson, 1978*; *Schouten et al., 2000*). Subsequent comparative genomics analyses have revealed the potential to produce sterol in a variety of bacterial groups including Planctomycetes, α-Proteobacteria, and Bacteroidetes (*Bode et al., 2003*; *Pearson et al., 2003*; *Wei et al., 2016*). Sterols produced by bacteria, however, tend to differ from eukaryotic sterols in both structure and in their biosynthetic pathways.

Sterol synthesis in both bacteria and eukaryotes requires the cyclization of the linear substrate oxidosqualene by an oxidosqualene cyclase (Osc) to generate either lanosterol or cycloartenol (*Abe, 2014*). However, fungi, vertebrates, and plants further modify these initial cyclization products to synthesize ergosterol, cholesterol, and stigmasterol, respectively (*Desmond and Gribaldo, 2009*). These biochemical transformations include demethylations, isomerizations, saturations, and desaturations that are essential for sterols to function properly in eukaryotes (*Nes et al., 1993*; *Xu et al., 2005*). Although bacterial production of fully modified sterols such as cholesterol is rare (*Lee et al., 2023*), some bacteria, including several aerobic methanotrophs and myxobacteria, do modify their sterols. For example, *M. capsulatus* produces sterols that are demethylated once at the C-4 and C-14 positions and contain a unique desaturation between C-8 and C-14 (*Bouvier et al., 1976*; *Figure 1A*). In addition, studies have revealed that bacterial proteins required to modify sterols can differ from the canonical eukaryotic sterol modifying proteins. Examples are the recently identified sterol demethylase proteins, SdmA and SdmB, in *M. capsulatus* that use $O_2$ to remove one methyl group at the C-4 position; the enzymes are mechanistically distinct from the eukaryotic C-4 demethylase enzymes (*Gachotte et al., 1998*; *Lee et al., 2018*; *Rahier, 2011*).

Although there is a better grasp of the taxonomic distribution of sterol synthesis in bacteria and of the distinct bacterial proteins involved in their biosynthesis, the function of these lipids in bacteria remains a mystery. It has been posited that bacterial sterols play a role in modulating the fluidic properties of the cytoplasmic membrane, similar to what is observed for cholesterol and ergosterol in eukaryotic cells (*Miao et al., 2002*; *Parks et al., 1995*; *Summons et al., 2006*). However, studies in *M. capsulatus* have demonstrated that the distribution of sterols differs considerably from what occurs in eukaryotes. Approximately 90% of the free sterol pool in eukaryotic cells is found in the cytoplasmic membrane (*Jacquier and Schneiter, 2012*) while in *M. capsulatus*, 75% of sterols are localized to the

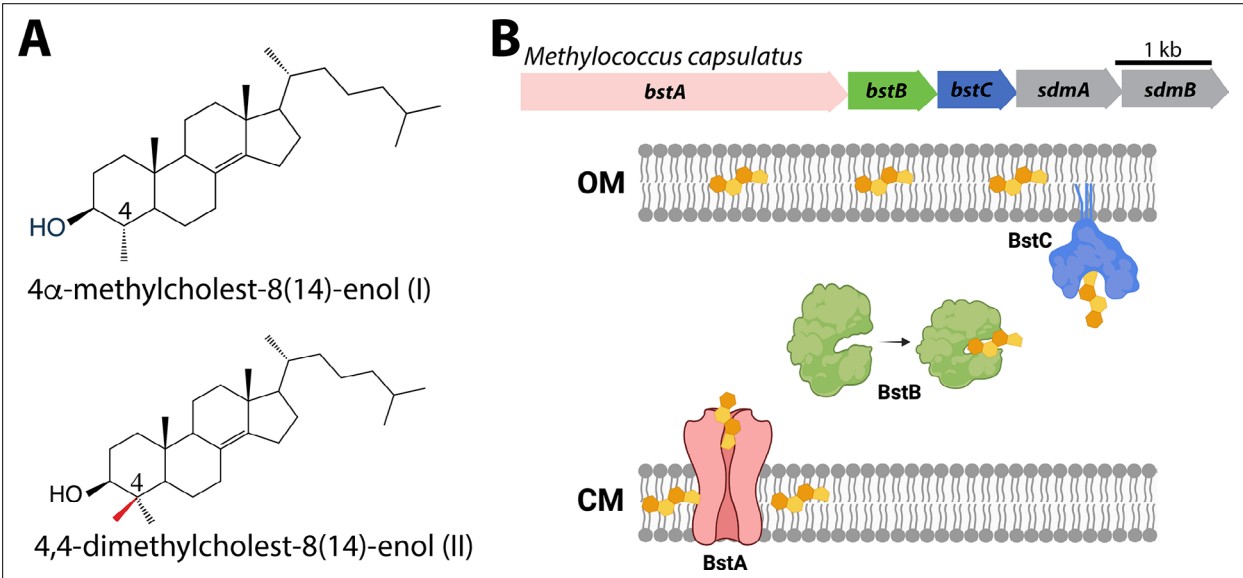

**Figure 1.** Sterols and their transport in *Methylococcus capsulatus*. (**A**) Structures of C-4 methylated sterols synthesized by *Methylococcus capsulatus*. (**B**) Operon of sterol biosynthesis and transport genes and (top) and schematic description of their localization. OM: outer membrane; CM: cytoplasmic membrane.

outer membrane (*Jahnke et al., 1992*). The outer membrane is the external asymmetric membrane of Gram-negative bacteria and differs from the cytoplasmic membrane in that its outer leaflet is composed of lipopolysaccharides (LPS) in addition to phospholipids (*Ruiz et al., 2006*). Additionally, sterols that differ by only a single methylation in the core structure can produce significantly different effects in terms of membrane fluidity and stability (*Bacia et al., 2005*). Given that *M. capsulatus* primarily produces sterols with one or two C-4 methyl groups, any sterol-membrane interactions in *M. capsulatus* could differ significantly from what is observed in eukaryotic membranes with cholesterol and ergosterol, which lack methyl groups at the C-4 position.

Interestingly, the 4,4-dimethyl sterols and 4-monomethyl sterols found in *M. capsulatus* also exist as precursors of pathway end-products in eukaryotes and accumulation of C-4 methylated sterols has been associated with a variety of eukaryotic processes. The accumulation of C-4 methylated intermediates is considered to be the cause of human genetic diseases known as sterolosis (*He et al., 2011*; *König et al., 2000*; *McLarren et al., 2010*) and 4,4-dimethyl sterols such as lanosterol in the brain are implicated in Parkinson's disease (*Lim et al., 2012*). In a variety of yeast species, C-4 methylated intermediates can regulate cellular processes associated with hypoxia and other conditions of cellular stress (*Hughes et al., 2007*; *Serratore et al., 2018*; *Todd et al., 2006*). Although the function of 4,4-dimethyl sterols and 4-monomethyl sterols in bacteria is still elusive, a functional role for these specific sterols beyond maintaining membrane fluidity and integrity seems plausible.

To better understand the significance of sterol utilization in the bacterial domain, a fuller understanding of the molecular mechanisms controlling sterol production and localization is needed. Given the observed distribution of sterols in the outer membrane of *M. capsulatus*, we hypothesized that transporters specific for C-4 methylated sterols must exist that can shuttle these substrates to the outer membrane. Trafficking of sterols to various organelles in eukaryotic cells is not fully understood (*Jacquier and Schneiter, 2012*) and impaired sterol transport is related to a variety of defects including lysosomal storage diseases (*Vance and Peake, 2011*). Identifying and characterizing bacterial proteins that transport sterol could reveal novel sterol binding motifs and folds and will provide meaningful insights into protein-sterol interactions and lipid trafficking more broadly.

In this study, we present three putative bacterial sterol transport proteins in *M. capsulatus* that exhibit remarkable specificity for C-4 methylated sterols. We first used bioinformatics to identify a cytoplasmic membrane protein (BstA), periplasmic protein (BstB), and outer membrane associated protein (BstC). Their ability to recognize and bind sterols was confirmed using protein-lipid pull down assays, where they showed selective binding to C-4 methylated sterols when in the presence of total lipid extracts from *M. capsulatus*. Quantitative assessment of ligand binding using Microscale Thermophoresis (MST) confirmed their preference for 4-monomethyl sterol, which they bind with equilibrium dissociation constants that are 30–90-fold lower than the 4,4-dimethyl sterol. High-resolution crystallographic structures of BstB and BstC reveal their pronounced divergence from eukaryotic sterol transporters. Docking studies and molecular dynamics simulations with sterol substrates reveal putative substrate binding sites and recognition mechanisms. Collectively, these data provide evidence for a novel system for sterol binding and possibly transport within the bacterial domain and advances our understanding of bacterial sterol lipids.

## Results

### Bioinformatic identification of sterol transport proteins in *M. capsulatus*

In a previous study, we used the Joint Genome Institute (JGI) Integrated Microbial Genomes (IMG) Phylogenetic Profiler to identify seventeen candidate genes in *M. capsulatus* that are unique to C-4 demethylating methanotrophs (*Lee et al., 2018*). Among them, a putative operon that contains five genes was found to localize next to three sterol biosynthesis genes. Two (*sdmA/sdmB*) have been characterized as demethylases involved in sterol demethylation at the C-4 position. Homologs of the other three hypothetical genes we have named *bstA, bstB* and *bstC* (bacterial sterol transporter), are found in all other aerobic methanotroph genomes that contain *sdmA/sdmB*. In some of these species, these three genes are also seen to be adjacent to other sterol synthesis genes such as the oxidosqualene cyclase (*osc*) and squalene monooxygenase (*smo*), implying they may be functionally related to sterol physiology (*Figure 1B*).

To obtain additional insights into the proteins encoded by these genes, we performed more refined bioinformatic analyses including protein sequence similarity network (SSN) (*Atkinson et al., 2009*) and structure prediction with Phyre2 (*Kelley et al., 2015*) and I-TASSER (*Yang et al., 2015*). BstA is predicted to belong to the Resistance Nodulation Division (RND) superfamily of transporters, which comprises inner membrane proteins that facilitate drug and heavy metal efflux, and protein secretion, amongst others. They are widespread in gram negative bacteria, but homologs are also found in archaea and eukaryotes. Interestingly, the human Newman Pick Type C1, a transporter of

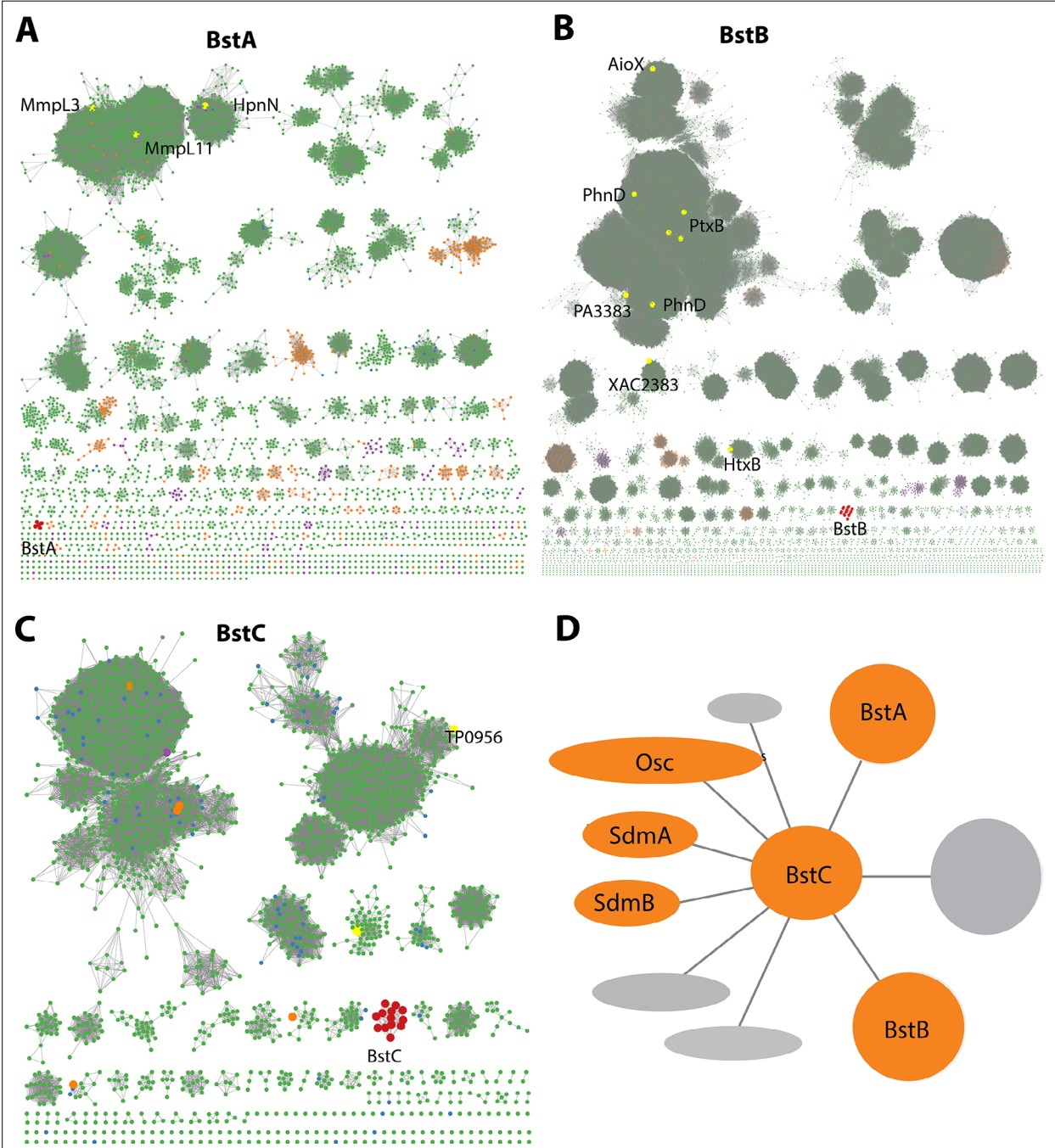

**Figure 2.** Bioinformatic analysis of Bst proteins. (**A–C**): Protein sequence similarity networks (SSN) of BstA, BstB and BstC with the cutoff of 35%, 40%, and 40% sequence identity, respectively. Taxonomic distribution of Bst protein families showing their presence in bacterial (green), eukaryotic (purple), and archaeal (orange) genomes. (**D**) Genome neighborhood network of Bst proteins. Shown in circles and ovals are genes within 10 genes of BstC; colored in orange are those with identified or predicted roles in sterol synthesis and trafficking.

sterols, also contains a transmembrane RND domain (*Nikaido, 2018*). The SSN of this superfamily shows that BstA is grouped into a single cluster containing five nodes (where sequences with >50% identity are merged into a single node). Structural prediction shows that BstA shares a high homology with a membrane-bound hopanoid transporter, HpnN (PDB 5KHS, *Burkholderia multivorans*), a RND transporter that moves hopanoids from the cytoplasm or cytoplasmic membrane (*Doughty et al., 2011*). In the SSN, HpnN is clustered separately from BstA, suggesting a possible functional divergence (*Figure 2A*, *Table 1*).

BstB belongs to a family of periplasmic/substrate binding proteins (SBPs/PBPs) known to traffic bacterial metabolites such as phosphonate. BstB proteins are grouped into a single cluster consisting of 16 sequences, with most of the sequences belonging to the *Methylococcaceae* family. Notably, the BstB proteins cluster separately even under stringent cutoffs of sequence identity (as low as 20%), hinting at a distinction of function compared with homologs (*Figure 2B*, *Table 1*). Structure prediction of BstB shows homology to PhnD (PDB 3QK6), a periplasmic solute-binding protein of the phosphonate uptake system in *E. coli* (*Alicea et al., 2011*). All structurally characterized PhnD homologs are distributed into different clusters in the SSN.

Finally, BstC is predicted to be a T-component member of the tripartite ATP-independent periplasmic component superfamily, which comprises periplasmic lipoproteins implicated in the cytoplasmic import of small molecules. The SSN of BstC reveals that the protein and its closest homologs are also grouped into a separate cluster even under a sequence identity cutoff of 20%. The cluster consists of 13 sequences, with most from *Methylococcaceae* and *Deltaproteobacteria* families. At the time of analysis, only one protein in the entire superfamily had structural or biochemical information: Tp0956 from *Treponema pallidum* (PDB 3U64) is found in the second largest cluster of the network (*Figure 2C*, *Table 1*). Compared with the SSNs of BstA and BstB, that of BstC shows the family is predominantly bacterial, with only 1 and 5 sequences from eukaryotes and archaea, respectively (*Figure 2A–C*).

In summary, sequence and predicted structural homology analyses reveal BstA, BstB, and BstC to be highly similar to transporters involved in disparate bacterial transport systems. However, their separation into distinct clusters in the SSNs imply their functions, including potential substrates, are not identical to their homologs. Because the Bst genes always co-localize with sterol synthesis genes (*osc*, *sdmA* and *sdmB*) in methanotrophs that produce sterols, we hypothesize that BstA/BstB/BstC represent a novel transport system for sterols in bacteria (*Figure 1B* and *Figure 2D*).

## Sterol interaction with transporters

To determine whether BstA, BstB, and BstC are sterol transporters, we first attempted gene deletion in *M. capsulatus* (data not shown). The deletion of these genes, along with others that are required for sterol synthesis (e.g. *osc*) were not successful, hinting that sterol synthesis and proper localization might be essential in this organism. Because these proteins share homology with well-known bacterial transporters, we reasoned that they are transporters and focused on determining their substrate preference by conducting pull-down assays to assess the binding of sterols from *M. capsulatus* lipid extracts. In these studies, an excess of recombinantly produced pure proteins was incubated with either the total lipid extract (TLE) from *M. capsulatus* or a specific polar fraction enriched for native hydroxy-lipids/sterols (HS). Using the HisPur Ni-NTA resin, proteins were isolated from the protein-lipid mixtures, and the protein-bound lipids were extracted and identified by GC-MS. In the presence of the TLE and HS fraction, BstA[PD] (the periplasmic domain of BstA; details in method), BstB, and BstC bound to 4-monomethyl and 4,4-dimethyl sterols (*Figure 3A*, *Figure 3—figure supplement 1*, *Table 2*). In contrast, neither the 4-monomethyl sterol, the 4,4-dimethyl sterol, nor any other lipids were present when the proteins were incubated with the DMSO negative control, suggesting that the recombinantly produced proteins did not co-purify with any lipids from the expression host. These results indicate that BstA[PD], BstB, and BstC preferentially bind C-4 methylated sterols in a mixture of native *M. capsulatus* sterols, hopanoids, and fatty acids.

## Quantitative analyses of protein-sterol interactions

The data from the pull-down assays demonstrated that C-4 methylated sterols have affinity for these putative sterol transporters. We next used microscale thermophoresis (MST) to determine the equilibrium binding affinities for these substrates. Binding curves were generated upon titration of the

**Table 1.** Available structural information in the SSN of different pfam members.

| Superfamily | Pfam | Protein | PDB | Organism |
|---|---|---|---|---|
| MMPL | PF03176 | HpnN | 5khs | *Burkholderia multivorans* |
| MMPL | PF03176 | MmpL3 | 6ajf | *Mycobacterium smegmatis* |
| MMPL | PF03176 | MmpL11 | 4y0l | *Mycobacterium tuberculosis* |
| Phosphonate-bd | PF12974 | AioX | 6esk | *Pseudorhizobium banfieldiae* |
| Phosphonate-bd | PF12974 | PhnD | 5lq5 | *Prochlorococcus marinus* |
| Phosphonate-bd | PF12974 | PtxB | 5lv1 | *Prochlorococcus marinus* |
| Phosphonate-bd | PF12974 | PtxB | 5o2k | *Pseudomonas stutzeri* |
| Phosphonate-bd | PF12974 | PtxB | 5jvb | *Trichodesmium erythraeum* |
| Phosphonate-bd | PF12974 | PA3383 | 3n5l | *Pseudomonas aeruginosa* |
| Phosphonate-bd | PF12974 | PhnD | 3qk6 | *Escherichia coli* |
| Phosphonate-bd | PF12974 | XAC2383 | 5ub3 | *Xanthomonas axonopodis pv. citri* |
| Phosphonate-bd | PF12974 | HtxB | 5me4 | *Pseudomonas stutzeri* |
| TatT | PF16811 | Tp0956 | 3u64 | *Treponema pallidum (strain Nichols)* |

labeled proteins with serially diluted sterol substrates, and the equilibrium dissociation constants were calculated by fitting the curves with different kinetic models (*Figure 3B*, methods). The '$K_d$' model applies to binding events with a single binding site or multiple independent binding sites, while the Hill model is invoked in instances where multiple binding sites that exhibit cooperativity are present, with Hill coefficients ($n_H$) greater than 1 signaling positive cooperativity. The average equilibrium dissociation ($K_d$) constant for BstA$^{PD}$ binding to 4-monomethyl sterol was determined to be 4.41±0.14 µM ($n_H$ = 4.0) and ≥203 ± 155 µM for the interaction with 4,4-dimethyl sterol. For BstB, the $K_d$ value of 2.43±0.13 µM ($n_H$ of 2.4) was determined for 4-monomethyl sterol; and ≥232 ± 84.3 µM for 4,4-dimethyl sterol. With BstC, the determined values were 2.50±0.39 µM and ≥80.9 ± 14.2 µM for 4-monomethyl sterol and 4,4-dimethyl sterol, respectively. All $K_d$ values for the 4,4-dimethyl sterol could not be accurately determined due to large errors caused by non-saturation even at the highest substrate concentration. In all three proteins, we measured clear binding to C-4 methylated sterols, with marked preference for the 4-monomethyl substrate. Additionally, BstA$^{PD}$ and BstB exhibit cooperativity for binding to 4-monomethyl substrate, indicating that multiple binding sites are plausible.

To further define the specificity of Bst proteins for sterols, we again used MST to detect their interactions with cholesterol and lanosterol, a lipid and precursor that differ in their methylation at the C-4 and C-14 positions, as well as their unsaturation patterns in the core ring structure. The MST results revealed no interaction between BstA$^{PD}$ and BstB with either cholesterol or lanosterol. However, BstC did bind to cholesterol with a $K_d$ value of 3.6±0.64 µM and to lanosterol with a value of 17.3±3.92 µM (*Figure 3—figure supplement 2*). These data hint that BstC is more tolerant to non-native substates.

## Crystallographic structure of BstB

To understand the molecular details that govern sterol recognition and binding, we crystallized BstB and obtained crystals of the apo form (*Table 3*). The 1.6 Å-resolution structure of was determined by experimental phasing, producing electron density that allowed the unambiguous building of a model of BstB (except for one disordered loop spanning residues 202–204). The structure comprises two globular α/β domains (*Figure 4A*, domains A and B) that form a cleft at the middle. Domain A contains six β-strands while domain B contains five. All the β-strands are flanked by αhelices. The domains are connected by two loops: one 9-residue loop that runs from β4 in domain A to β5 in domain B, and a second 8-residue loop that connects β9 in domain B and β10 in domain A. These two long loops could work as a hinge to allow a bending motion of two domains to induce a conformational change upon substrate binding; this is often observed with SBPs/PBPs (*de Boer et al., 2019*). In addition, an α-helix formed by the C-terminal residues lies behind the cleft and may also allow flexibility of the two domains. Several extend loops are found on the domain-domain interface. The loops (residues 34–38

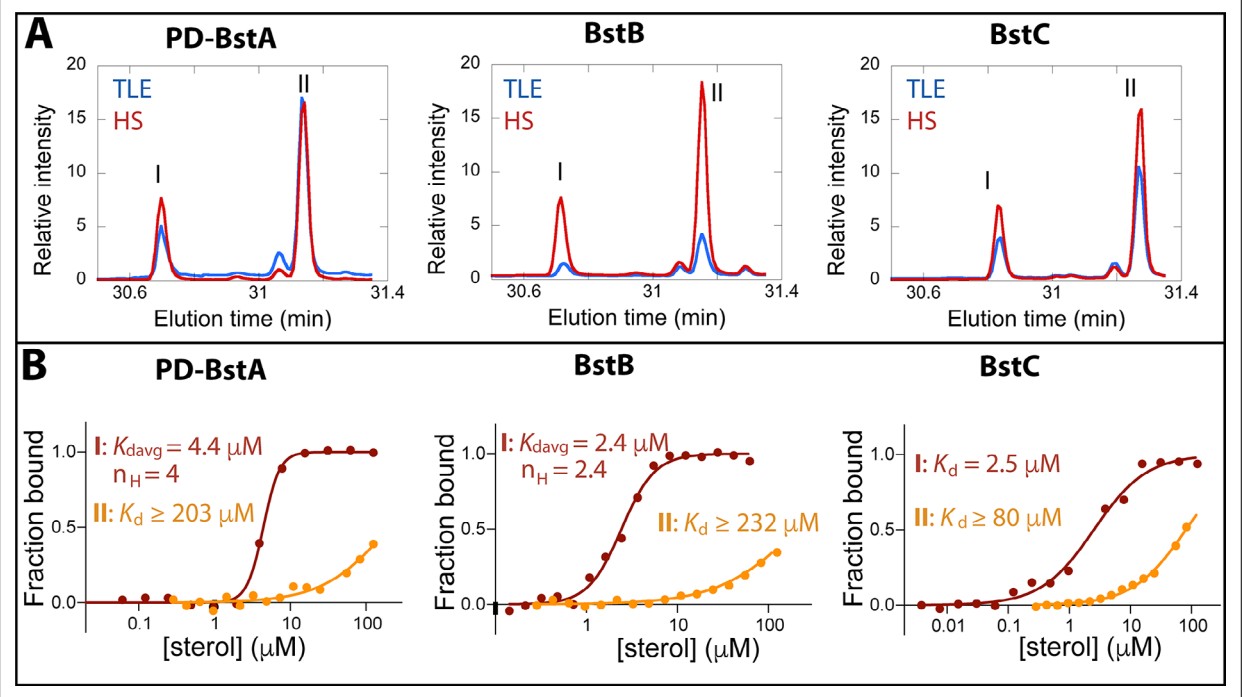

**Figure 3.** Binding of C-4 methylated sterols to transporter proteins. (**A**) GC chromatograms of sterols bound to purified (40 µM) *M. capsulatus* BstA[PD], BstB, and BstC during protein-lipid pull-down assays. The proteins preferentially bind 4-monomethyl (I) and 4,4-dimethyl (II) sterols when incubated with the total lipid extract (TLE) and hydroxy lipid (HS) fraction from *M. capsulatus*. (**B**) MST analysis of the proteins (50 nM) binding to C-4 methylated sterols (3.8 nM –1125 mM). Equilibrium dissociation constants ($K_d$) and Hill coefficients ($n_H$) are reported on the plots.

The online version of this article includes the following source data and figure supplement(s) for figure 3:

**Figure supplement 1.** Purified BstABC proteins.

**Figure supplement 1—source data 1.** Raw purification gel of PD-BstA.

**Figure supplement 1—source data 2.** Marked purification gel of PD-BstA.

**Figure supplement 1—source data 3.** Raw purification gel of BstB.

**Figure supplement 1—source data 4.** Marked purification gel of BstB.

**Figure supplement 1—source data 5.** Raw purification gel of BstC.

**Figure supplement 1—source data 6.** Marked purification gel of BstC.

**Figure supplement 1—source data 7.** Raw pulldown gel of PD-BstA.

**Figure supplement 1—source data 8.** Raw pulldown gel of PD-BstA.

**Figure supplement 1—source data 9.** Raw pulldown gel of BstB.

**Figure supplement 1—source data 10.** Raw pulldown gel of BstB.

**Figure supplement 1—source data 11.** Raw pulldown gel of BstC.

**Figure supplement 1—source data 12.** Raw pulldown gel of BstC.

**Figure supplement 2.** Binding of cholesterol (top) and lanosterol (bottom) to BstA[PD], BstB, and BstC.

and residues 171–177) exhibit poor electron densities, suggesting conformational sampling of more than one state (*Figure 4—figure supplement 1A*).

The cleft between domains A and B forms a bulky cavity with an area of ~1179.5 Å² and a volume of ~1009.6 Å³ that extends into domain A and is only accessible to solvent from one side of the structure. The cavity is predominantly hydrophobic, suggesting large and hydrophobic substrates like sterols are favorable for binding (*Figure 4B*, *Table 4*). Additionally, a second pocket (area of ~172 Å²; volume of ~117 Å³) is found in domain B and close to the central cleft. This pocket is localized behind the unmodeled loop (residues 202–204), with the extremely poor density in this part implying high flexibility. This pocket is highly hydrophobic, but its role (if any) in ligand binding is unexplored (*Figure 4B*; *Figure 4—figure supplement 2*). Other separate hydrophobic areas are also observed

**Table 2.** Protein sequences and primers.

| NAME | SEQUENCE | PRIMERS |
|---|---|---|
| BstA | MNIPHLAALAAERFARRPWRVLALAMALSALSL WAVSRLPVHTSRQALLPHDNAVAQRFDAFLDK FGAASDLIVVLEGAPPDELKPFADELATALAAEP EIAQATARLDLRFVLEHAYLAVTPERLGTLAGVL EKFGAGAIPEDSSQVDATLGRLLQWLEGAPAM PAAGIDLPTVEVGLKLLGASLDEWHRWLSAGE VPAALDWTRLLAGLGGSEIANDGYFVSRDGRM YFLFVHPASASEDFTAIGPFVEKVRTVAADRAAR ARAAGRTAPKVGLTGLPAIEYEEHVSIRHDIALVV GSAAGLIVLLILVVVRSWRWALVIFVPMGLGVLW SLGLALVTIGHLTLITASFIAVLFGLGADYGIFTSAR IAEERRRGKPLTEAIGAGMGASFQAVFTAGGASV VIFGALATVDFPGFSELGLVAAKGVMLILVSTWLV QPALYALLPPKLAPLPAAASAGAIEPGRMPFRGS VAVILVAGALATAAFGIGSGYELPFDYDVLSLLPK DSESAYYQNRMVAESDYQAEVVIFTAPDLEEAR RIAAEAGRLGSVAKVQSLMDLFPPDADARALEA RRIGELADDGYAVRLARLAAIGLPEGTFGRVRTI LEKGGDFIDQSQELAFSAGHSGLVAALEDVRGR LDAVRSAIEADPVQARERSERFFRMLLSAAERG VALLAEWRQARPITPAQLPPALRDRFFAADGTVA VYAFPAKTVYDPANLDRLMQEIYGVSPDATGFPA THQVFSKSVVESFTHGTREAVTVCLLWLALVLRN WRGFVLASMPLLIGGGWMLGLMALCGIRYNYAN IIALPLVIALAVDYGVWFSQRWFDLKDRSLTQINRV AGGVIGLAAGTELAGLGAITLANYRGVSSLGVNI TVGLLCCLAATLWVAPAIGQLLDSRKKP | pET20b-Forward: GAAGGAGATATACATATGAACATCCCCCATTTGGCCGCTC pET20b-Reverse: GTGGTGGTGGTGCTCGAGTGGTTTCTTTCTTGAATCGAGAAGC pET28a-Forward: AGAATCTTTATTTTCAGGGCCATAACATTCCGCACCTGGCGGC pET28a-Reverse: CTCAGTGGTGGTGGTGGTGGTGCTCGAGCTTAGGTTTCTTACGGCTGTCCAG |
| BstA^PD | MAVAQRFDAFLDKFGAASDLIVVLEGAPPDELKP FADELATALAAEPEIAQATARLDLRFVLEHAYLAV TPERLGTLAGVLEKFGAGAIPEDSSQVDATLGRL LQWLEGAPAMPAAGIDLPTVEVGLKLLGASLDE WHRWLSAGEVPAALDWTRLLAGLGGSEIANDG YFVSRDGRMYFLFVHPASASEDFTAIGPFVEKVR TVAADRAARARAAGRTAPKVGLT**GSGSGSGS**ES AYYQNRMVAESDYQAEVVIFTAPDLEEARRIAAE AGRLGSVAKVQSLMDLFPPDADARALEARRIGE LADDGYAVRLARLAAIGLPEGTFGRVRTILEKGG DFIDQSQELAFSAGHSGLVAALEDVRGRLDAVR SAIEADPVQARERSERFFRMLLSAAERGVALLA EWRQARPITPAQLPPALRDRFFAADGTVAVYAF PAKTVYDPANLDRLMQEIYGV | pET28a-Forward1: CTTTATTTTCAGGGCCATATGGCCGGTGGCGCAACGTTTCGAC pET28a-Reverse1: AGTACGCGCTCTCGCTACCGCTACCGCTGCCGCTACCGGTCAGACCAACTTT CGGC pET28a-Forward2: GTTGGTCTGACCGGTAGCGGCAGCGGTAGCGGTAGCGAGAGCGCGTACTATC AGAAC pET28a-Reverse2: CTCAGTGGTGGTGGTGGTGGTGCTCGAGTCACACACCATAAATTTCCTG |
| BstB | M**TDKITCFSLLAALLAPFVPAQA**GAPAPVVVCY PGGAVNERDADQAMDAMLRVVERVGQWPEKS FSSVFTAKVADCGKLMAEMKPAFAITSLALYLDM RGQYDLVPVVQPRIDGRTSERYRVVAKQGRFH DMDELKGRTLGGTMLDEPAFLGKIVFAGKYDPE KDFALQPSRQAIRALRSLDKGELDAVVLNEQQF AGLSALQLASPVETVFTSAEIPLMGVVANARLTS AQERARFAQALETLCADPEGRKLCDLFGIQSFV AVDPTVFDPMARLWLARN | pET20b-Forward: GAAGGAGATATACATATGGGCGCCCCCGCACCG pET20b-Reverse: GTGGTGGTGGTGCTCGAGGTTCCGCGCCAGCCA pET28a-Forward: CGAGAATCTTTATTTTCAGGGCCATGGCGCCCCCGCACCGGTGGTC pET28a-Reverse: CTCAGTGGTGGTGGTGGTGGTGCTCGAGTTAGTTCCGCGCCAGCCAGAG |
| BstC | M**RLFAAGILAGVLAG**CGGLHRDGTPAGPSAGC PRLTAAALSAGQDALGPSSETQELECALDFLRG SDDPALRRSSLGSRICLHLAERNSDPAERARFA REGVERAEAALAQGGEDDGAVHYYLAANLGLA VRDDMTAALANLHRLEHESEAAVKLSPDFDDG GPLRLLGMLYLKAPAWPAGMGDGDKALDLLGQ AVERHPGHPLNHLFYAEALWEVNGESESRRVE EEMAAGWRLLESGSWGYNKQIWKREFADLRQ EIGAPAR | pET20b-Forward: GAAGGAGATATACATATGTGCGGCGGTCTTCAC pET20b-Reverse: GTGGTGGTGGTGCTCGAGCCTAGCGGGCGCCCC pET28a-Forward: ATCTTTATTTTCAGGGCCATTGCGGCGGTCTTCACCGCGATG pET28a-Reverse: CTCAGTGGTGGTGGTGGTGGTGCTCGAGTTACCTAGCGGGCGCCCCGAT |

The signal peptides in BstB and BstC are shown in bold.

on the protein surface and a new hydrophobic interface could be formed by two hydrophobic areas across the cleft upon conformational change induced by substrate binding (*Figure 4—figure supplement 2*). These areas may be helpful for mediating the protein-protein interactions with BstA or BstC to facilitate the substrate transfer.

A DALI structural homology search (*Holm and Rosenström, 2010*) suggests the structure of BstB closely resembles that of PBPs that share a bi-lobe architecture and undergo conformational change

**Table 3.** Data collection and refinement statistics.

| | apo-BstB(SeMet) | apo-BstC(SeMet) |
|---|---|---|
| Resolution range | 38.4–1.6 (1.657–1.6) | 28.63–1.91 (1.978–1.91) |
| Space group | P 21 21 21 | P 1 21 1 |
| Unit cell (a, b, c; a, b, g) | 39.882, 40.355, 142.263; 90, 90, 90 | 55.083, 57.257, 85.825; 90, 97.26, 90 |
| Total reflections | 606476 (60185) | 200826 (19731) |
| Unique reflections | 31071 (3061) | 41329 (4084) |
| Multiplicity | 19.5 (19.7) | 4.9 (4.8) |
| Completeness (%) | 97.81 (93.74) | 99.91 (99.98) |
| Mean I/sigma(I) | 30.96 (2.07) | 11.09 (2.78) |
| Wilson B-factor | 22.86 | 15.59 |
| R-merge | 0.07677 (1.462) | 0.1267 (0.6131) |
| R-meas | 0.07881 (1.501) | 0.1423 (0.6891) |
| R-pim | 0.01755 (0.3345) | 0.06419 (0.3117) |
| CC1/2 | 1 (0.745) | 0.994 (0.762) |
| CC* | 1 (0.924) | 0.999 (0.93) |
| Reflections used in refinement | 30,554 (2873) | 41,317 (4084) |
| Reflections used for R-free | 1964 (184) | 1996 (198) |
| R-work | 0.2038 (0.2892) | 0.174 (0.2389) |
| R-free | 0.2234 (0.2749) | 0.207 (0.2675) |
| CC(work) | 0.955 (0.786) | 0.948 (0.879) |
| CC(free) | 0.948 (0.725) | 0.939 (0.834) |
| Number of non-hydrogen atoms | 2117 | 4036 |
| macromolecules | 1974 | 3546 |
| solvent | 142 | 355 |
| Protein residues | 257 | 464 |
| RMS(bonds) | 0.010 | 0.012 |
| RMS(angles) | 1.03 | 1.28 |
| Ramachandran favored (%) | 99.21 | 100.00 |
| Ramachandran allowed (%) | 0.79 | 0.00 |
| Ramachandran outliers (%) | 0.00 | 0.00 |
| Rotamer outliers (%) | 0.97 | 0.00 |
| Clashscore | 2.28 | 1.39 |
| Average B-factor | 33.82 | 19.33 |
| macromolecules | 33.84 | 18.22 |
| ligands | 30.00 | 27.28 |
| solvent | 33.49 | 27.44 |
| Number of TLS groups | 3 | 1 |

Statistics for the highest-resolution shell are shown in parentheses.

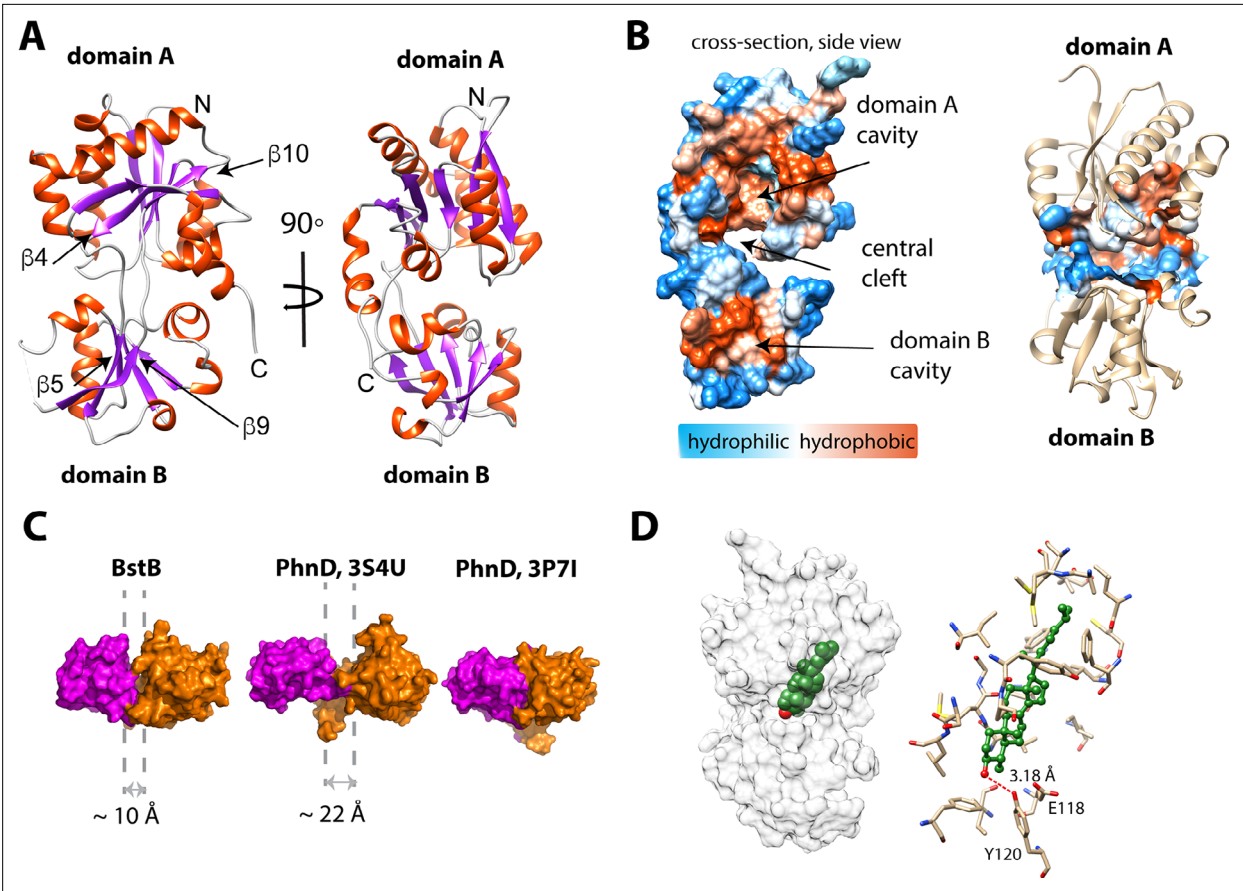

**Figure 4.** Structure of BstB. (**A**) Cartoon representation of BstB with the α-helixes and β-strands colored in red and magenta, respectively. (**B**) Hydrophobicity representations of BstB show a cleft in the middle and a cavity in domain A. The cavity is dominatingly hydrophobic and open only to one side. (**C**) Comparison of the cleft between the two domains of BstB, apo-PhnD (3S4U), and liganded PhnD (3P7I). BstB displays a similar but narrower opening (~10 Å) compared to non-ligand PhnD (~22 Å). (**D**) Docking model of the structures of BstB with 4-monomethyl sterol. Left, the docking model is shown in surface representation with 80% transparency. Sterol is shown in sphere and colored in green with the hydroxyl group is colored in red; Right, docking complex of $Mono_A$. The surrounding residues are shown in stick and ball. Hydrogen bonds between the hydroxyl group of the sterol and Tyr120 is shown in the red dashed line.

The online version of this article includes the following figure supplement(s) for figure 4:

**Figure supplement 1.** A portion of the 2Fo − Fc electron density omit map contoured at 2 σ and the Fo−Fc electron density omit map contoured at 3σ (green mesh) of BstB.

**Figure supplement 2.** Hydrophobicity representation of the BstB structure shows a second hydrophobic pocket located in domain B below the missing loop (202-204).

**Figure supplement 3.** Docking models of 4-monomethyl sterol and 4,4-dimethyl sterol in BstB structure.

**Figure supplement 4.** MST binding curves of BstB variants with 4-monomethyl sterol.

**Figure supplement 5.** MD simulation of apo-BstB and sterol-docked BstB.

upon substrate binding to the central cavity (*Quiocho and Ledvina, 1996*). A comparison of the BstB structure with the phosphonate-binding protein PhnD from *E. coli* with and without its substrate (2-aminoethyl phosphonate, 2AEP, PDB 3P7I and 3S4U) was performed (*Alicea et al., 2011*). The comparison revealed that BstB adopts an intermediate conformation between the unliganded PhnD (3S4U) and liganded one (3P7I, *Figure 4C*). The cleft in BstB is less open (~10 Å) than that in apo-PhnD (~22 Å). There are several possible explanations for this: (1) compared with apo-PhnD structure, the cleft of BstB is deeper and more hydrophobic. Given that BstB binds to sterols (which have a nearly planar architecture and are more hydrophobic than phosphonates), the different architecture of the cleft in BstB may be related to its functional role; (2) it is known that binding of non-substrate ligands can at times trigger a conformational change in PBPs (*de Boer et al., 2019*). Although no obvious

**Table 4.** Residues involved in forming the cavity of BstB.

| Domain | SeqID | AA | Domain | SeqID | AA |
|---|---|---|---|---|---|
| A | 29 | Val | A | 115 | Arg |
| A | 31 | Val | A | 117 | Ser |
| A | 33 | Tyr | A | 118 | Glu |
| A | 34 | Pro | B | 120 | Tyr |
| A | 35 | Gly | B | 143 | Thr |
| A | 36 | Gly | B | 144 | Met |
| A | 37 | Ala | B | 146 | Asp |
| A | 38 | Val | B | 147 | Glu |
| A | 39 | Asn | B | 150 | Phe |
| A | 40 | Glu | B | 171 | Ser |
| A | 43 | Ala | B | 172 | Arg |
| A | 44 | Asp | B | 173 | Gln |
| A | 46 | Ala | B | 174 | Ala |
| A | 47 | Met | B | 175 | Ile |
| A | 48 | Asp | B | 192 | Asn |
| A | 49 | Ala | B | 193 | Glu |
| A | 50 | Met | B | 194 | Gln |
| A | 51 | Leu | B | 195 | Gln |
| A | 53 | Val | B | 217 | Ile |
| A | 54 | Val | A | 218 | Pro |
| A | 67 | Ser | A | 219 | Leu |
| A | 69 | Phe | A | 221 | Gly |
| A | 71 | Ala | A | 222 | Val |
| A | 73 | Val | A | 223 | Val |
| A | 89 | Ile | A | 237 | phe |
| A | 90 | Thr | A | 241 | Leu |
| A | 91 | Ser | A | 244 | Leu |
| A | 92 | Leu | A | 253 | Leu |
| A | 93 | Ala | A | 254 | Cys |
| A | 94 | Leu | A | 256 | Leu |
| A | 95 | Tyr | A | 257 | Phe |
| A | 106 | Pro | A | 258 | Gly |
| A | 108 | Val | A | 259 | Ile |
| A | 109 | Gln | A | 262 | Phe |
| A | 110 | Pro | A | 270 | Phe |
| A | 112 | Ile | | | |

ligand density is observed in the BstB structure, weak unmodeled densities were observed inside the central cavity. Most of these are small patches that cannot be satisfactorily modeled with water and do not closely correspond to any reagent used for crystallization. One of these patches in the central cleft could be modeled with a chloride ion engaging in polar interactions with the surrounding Glu118, Tyr120, and Phe150. Additionally, more unmodeled positive density is observed in further into the cavity in domain A (*Figure 4—figure supplement 1B–C*). It is plausible that the binding of ligands co-purified with the protein triggered a conformational change leading to a 'partially-open' state. A third possibility is that BstB is structurally distinct from the PhnD homologs in that its cleft is much smaller.

## Substrate docking and molecular dynamics simulation of BstB

To obtain greater insights into the mechanism of substrate recognition and binding in the absence of a substrate-bound structure, we used ICM-Pro (*Abagyan et al., 1994b*; *Abagyan and Totrov, 1994a*) to generate docking models of C-4 methylated sterol binding to BstB. A three-dimensional model of 4α-monomethyl sterol was constructed from the structure of cholesterol using the Molecular Editor in ICM-Pro and then docked to the BstB structure. In the first iteration, the protein was held rigid, and the ligand allowed to dock flexibly. In this docking model, the sterol was positioned in the hydrophobic cavity in domain A with the polar head pointing toward the central cleft and domain B, adjacent to three residues: Glu118, Tyr120, and Asn192. In subsequent replicate docking iterations, the side chains of these three residues were allowed to be flexible along with the ligand. There are three energetically favored poses: two ($Mono_A$ and $Mono_B$) form a single hydrogen bond to Tyr120, while pose $Mono_C$ where the hydroxyl makes two hydrogen bonds to Tyr120 and Asn192 (*Figure 4D* and *Figure 4—figure supplement 3A–C*, *Table 5*).

To probe the dynamics of the BstB structure during substrate binding, we performed multiple 20 ns molecular dynamics (MD) simulations using Desmond (Schrodinger) on each docked poses for monomethyl sterol from ICM-Pro. In all cases, analysis of the resultant MD trajectories indicated that the simulations became stable after the first few picoseconds. In the three poses where there was initially an interaction with Tyr120, that interaction is lost and replaced with a new hydrogen bond with Glu118, which persists for 85–95% of

**Table 5.** Docking and molecular dynamics results for BstB.

| | | ICM docking results | | | | Hydrogen bonds in docked poses | | | Hydrogen bonds after MD | | |
| | | energies (kcal/mol)* | | | | distance (Å) | | | bond persistence (%) | | |
| Ligand | Pose | Score† | Hbond | Hphob | VwInt | Glu118 | Tyr120 | Asn192 | Glu118 | Tyr120 | Asn192 |
|---|---|---|---|---|---|---|---|---|---|---|---|
| | A | −17.7 | −0.94 | −9.62 | −25.60 | - | 3.18 | - | 95 | 3 | - |
| | B | −20.4 | −0.69 | −9.50 | −26.10 | - | 3.49 ‡ | - | 95 | 2 | 1 |
| Mono | C | −16.9 | −2.77 | −9.32 | −23.73 | - | 2.83 | 3.17 | 85 | - | 11 |
| | A | −20.3 | −4.53 | −9.48 | −23.22 | 2.94 | 2.74 | 2.89 | 12 | 12 | 57 |
| | B | −27.8 | −4.64 | −9.74 | −28.74 | 2.84 | 2.83 | - | 94 | 10 | 1 |
| Di | C | −17.2 | −1.99 | −9.94 | −24.13 | - | 3.18 | - | - | - § | - |

*Energies for hydrogen bonding (Hbond), hydrophobic interactions (Hphob) and van der Waals interactions (VwInt).

†The score has no units. More negative numbers are indicative of stronger binding.

‡This distance could be considered too long to be a hydrogen bond.

§This pose loses the hydrogen bond with Tyr120 but gains a hydrogen bond with the carbonyl oxygen of Pro34 which then persists for 95% of the MD simulation.

the simulation. Pose Mono_C also has an intermittent interaction with Asn192 for approximately 11% of the simulation. In all simulations, whether there is a hydrogen bonding interaction with the ligand or not, the amide group of Asn192 is flipped.

To better probe the protein's substrate preference, docking models and MD simulations with the 4,4-dimethy sterol were also generated. In these models, the 4,4-dimethyl sterol also docks to the cavity in domain A with three poses of roughly equal energies (*Figure 4—figure supplement 3D–F*, *Table 5*). In one (pose Di_A), hydrogen bonds to Glu118, Tyr120 and Asn192 are possible; in a second (Di_B), hydrogen bonds to Glu118 and Tyr120 are possible; the third (Di_C) has a hydrogen bond to Tyr120 only. The 4,4-dimethyl sterol showed interesting behavior during the 20 ns simulations. Pose Di_A, which had three hydrogen bonds after ICM-Pro docking, retained all three interactions (although those with Glu118 and Tyr120 are intermittent and persist for only 12% of the simulation). Pose Di_B retained both interactions from the docking model, although the hydrogen bond to Tyr120 was only present for 10% of the simulation. The third pose, Di_C, showed the largest structural changes, losing the one interaction it had with Tyr120. In this pose, the head group of the sterol moved out of the binding pocket delineated by Glu118, Tyr120 and Asn192 and a new hydrogen bond with the carbonyl oxygen of Pro34 formed, persisting for the duration of the simulation (*Table 5*). It is possible that these differences, although subtle, explain the reduction in affinity for the dimethyl sterol substrate.

The aliphatic tails of all docked sterols reside in a hydrophobic cavity in domain A lined by the side chains of Tyr33, Pro34, Ala46, Met47, Met50, Ile89, Ser91, Leu92, Pro110, Ile112, Leu219, Met220, Val222, Leu244, Leu253, Cys254, Phe257, Ile259, and Phe262 in domain A. Sequence alignment shows that these residues are highly conserved between proteins in the same SSN cluster, implying their significance in forming the substrate binding pocket. Mutagenesis to remove or reduce the potential interactions with the polar head and the aliphatic tail showed subtle influences (<twofold) on the sterol binding affinity, indicating that the interaction between the protein and sterol is extensive and principally depends on hydrophobic interactions that are not easily disrupted (*Figure 4—figure supplement 4*). Moreover, the conformational dynamics of PBPs could mask the identity of additional residues that might be involved substrate binding and stabilization. Regardless, the determination of the sterol binding site in BstB remains to be experimentally verified.

To investigate whether BstB can undergo conformational change (as most PBPs do upon substrate binding), a 150 ns molecular dynamics simulation on apo-BstB was performed (*Video 1*). Plotting the *RMSD* of the simulated structure

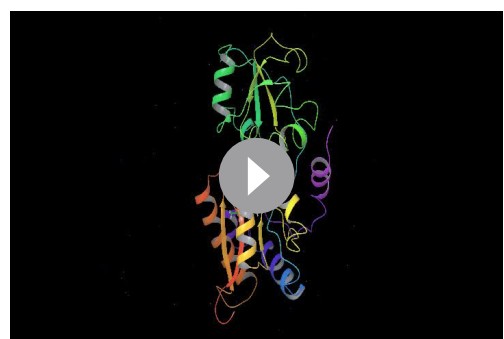

**Video 1.** 150 ns molecular dynamics simulation of the apo-BstB structure.

https://elifesciences.org/articles/90696/figures#video1

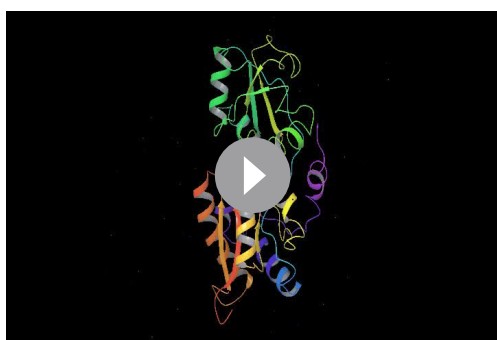

**Video 2.** 150 ns molecular dynamics simulation of the *mono$_c$*-docked BstB complex.
https://elifesciences.org/articles/90696/figures#video2

against the initial crystal structure at 25 ps intervals during the MD trajectory shows that for the first ~90 ns the protein showed some degree of flexibility (*Figure 4—figure supplement 5A*). During this part of the trajectory, the *RMSD* oscillates between 2.0 and 3.1 Å (average RMSD = 2.6 Å). At ~90 ns, there was a distinct conformational change in the structure giving rise to conformations with *RMSD*s ranging from 2.7 to 4.5 (average RMSD = 3.3 Å). Visual analysis of the MD trajectory shows that at ~90 ns the two domains open by approximately 20° (*Figure 4—figure supplement 5B*, *Video 1*). In contrast, the simulation (150 ns) on the *Mono$_C$* docked complex which had a *RMSD* of 2.6 Å for the entire length of the trajectory and did not show the same conformational change as the apo-BstB simulation, which supports the idea that the apo-BstB structure represents a 'partially-open' form of the protein (*Video 2*).

## Crystallographic structure of BstC

The structure of BstC was also determined by experimental phasing to a resolution of 1.9 Å. Two monomers (*RMSD* of 0.656 Å for all Cαs) in the asymmetric unit form a dimer with 180 rotational symmetry. Each monomer adopts an all-α-helical structure with 12 α-helices in total, designated with H1 to H12; 9 antiparallel arranged helices (H4-H12) form an irregular ring with a pore running through the center. The H4, H6, H8, H10, H12 constitute the inside wall of the pore, while the others form the outside wall. Two helices (H1-H2) at the N-terminus sit on the side-top of the ring with a short helical turn (H3) connecting the two (*Figure 5A*).

Hydrophobicity analysis shows that BstC has a predominantly hydrophilic exterior surface (*Figure 5—figure supplement 1A*). Three cavities are found in the center of the structure. We annotated them as Tunnel 1 (TN1), Cavity 1 (CA1) and Cavity 2 (CA2), with total area of ~597.1 Å$^2$ and total volume of ~447 Å$^3$. TN1 is closer to the N-terminal face of the structure, while CA1 and CA2 are proximal to the C-terminal face and form two open hydrophobic pockets; TN1 exhibits a mixture of hydrophobic and hydrophilic amino acids (*Figure 5B* and *Figure 5—figure supplement 1B*, *Table 6*). The extensive hydrophobic environment inside the BstC structure suggests it can accommodate large hydrophobic substrates. Notably, electrostatic calculation shows that most of the surface of BstC, especially the C-terminal face, exhibits strong negative charge (*Figure 5—figure supplement 2*). These charged surfaces may mediate protein-protein interactions, perhaps with BstB.

A structure similarity search using the DALI server revealed high homology to Tp0956 (PDB 3U64, *Treponema pallidum strain Nichols*). Tp0956 is the T-component protein of a subfamily of TRAP (tripartite ATP-independent periplasmic) transporters known as TPATs (tetratricopeptide repeat-protein associated TRAP transporters), which are implicated in the transport of organic acid ligands in bacteria (*Brautigam et al., 2012*). The true physiological substrate of Tp0956 has not been determined, but *Treponema* is a spirochete known to rely on import of host lipids and fatty acids to construct its cell envelope (*Radolf et al., 2016*).Tp0956 contains four helical hairpins that are similar to tetratricopeptide repeat (TPR) motifs, which are hallmarks of proteins that are involved in protein-protein interactions (*D'Andrea and Regan, 2003*). Comparison of the structures show that two antiparallel α-helices exist in Tp0956 are missing in the structure of BstC (between H5-H6), as well as a shorter H4 and H5. But the TPR-like substructure is also observed in BstC and forms part of the pore and lateral surface of the protein, hinting that BstC might engage with itself or other proteins (*Figure 5—figure supplement 3*). Size-exclusion chromatography coupled with multiangle light scattering (SEC-MALS) analysis shows that indeed the purified BstC exists both as a monomer and a dimer in solution with a ratio ~3:1 (*Figure 5—figure supplement 4*). However, the role of the dimerization in this protein is not clear and its determination will require additional studies.

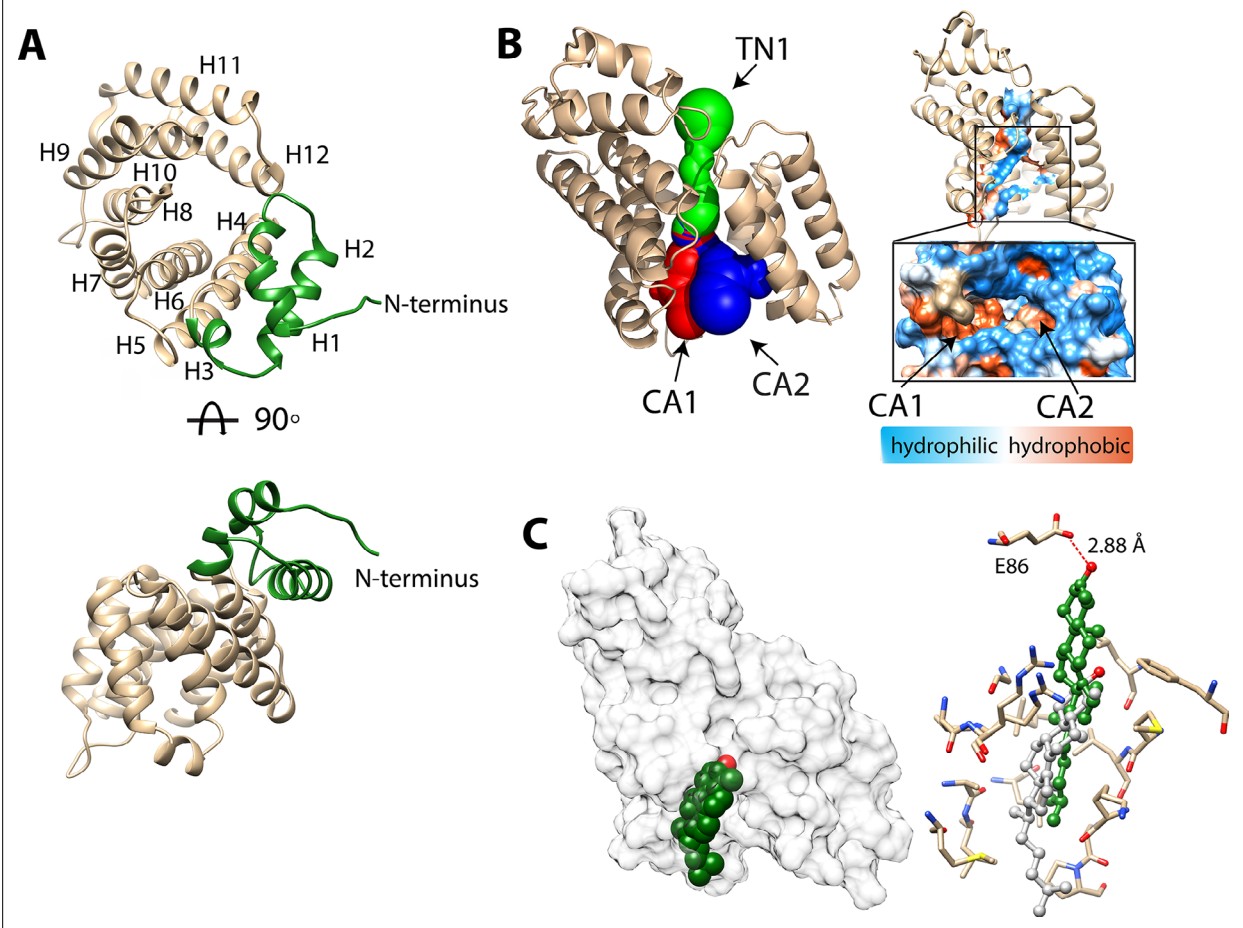

**Figure 5.** Structure of BstC. (**A**) Cartoon representation of BstC. The two α-helixes at the N-terminus are colored in green. Two monomers (RMSD of 0.656 Å for all Cαs) with 180 rotational symmetry exist in the asymmetric unit. (**B**) A representation of cavities BstC (left, with Tunnel 1 colored in green, Cavity 1 in red, and Cavity 2 in blue) alongside hydrophobicity representation of the cavities (right). (**C**) Docking model of the structures of BstC with 4-monomethyl sterol. Left, the docking model is shown in surface representation with 80% transparency. Sterol is shown in sphere and colored in green with the hydroxyl group is colored in red; Right, docking complex before and after MD simulation. The surrounding residues are shown in stick and ball. The sterol prior to the MD simulation is colored in grey and while green shows its position after the simulation. A hydrogen bond between the hydroxyl group of the sterol and Glu86 is shown as a red dashed line.

The online version of this article includes the following figure supplement(s) for figure 5:

**Figure supplement 1.** Hydrophobicity representations of the BstC structure on (**A**) the exterior surface and (**B**) the cross-section of the side view of the tunnel in BstC structure.

**Figure supplement 2.** Electrostatic calculation shows that most of the surface, including the N-terminal (**A**) and C-terminal (**B**) terminal faces, exhibits strong negative charge.

**Figure supplement 3.** Structural comparison of BstC and Tp0956.

**Figure supplement 4.** Size exclusion chromatography (SEC, left) and SEC-multiangle light scattering (MALS, middle and right) analysis of BstC.

**Figure supplement 5.** Docking models of 4-monomethyl sterol and 4,4-dimethyl sterol in BstC structure.

**Figure supplement 6.** MST analysis of the BstC variants (50 nM) binding to sterols.

## Substrate docking and molecular dynamics simulations with BstC

To assess substrate-binding, we performed the docking of 4-monomethyl sterol and 4,4-dimethyl sterol into the BstC structure using ICM-Pro. The final docking model was obtained by choosing the one with the most negative ICM-Pro score. All the sterols were found embedded in the hydrophobic cavity (CA1) lined by Gly128, Val131, Arg132, Met135, Ala138, Leu139, Leu142, Leu145, Leu167, Met170, Leu171, Lys174, Ala175, Pro176, and Phe208, with the hydroxyl group pointing inwards and the aliphatic tail pointed out (*Figure 5C* and *Figure 5—figure supplement 5A–B*). Residues involved

**Table 6.** Residues involved in forming the hydrophobic cavity in BstC.

| Tunnel1 | Chamber2 | Chamber3 |
|---------|----------|----------|
| ASP 45  | GLY 128  | LEU 129  |
| PRO 49  | LEU 129  | ARG 132  |
| LEU 82  | ARG 132  | LEU 167  |
| HIS 83  | MET 135  | MSE 170  |
| GLU 86  | ALA 138  | LYS 174  |
| TYR 121 | LEU 139  | LEU 207  |
| TYR 122 | LEU 142  | PHE 208  |
| LEU 129 | LEU 145  | GLU 211  |
| ARG 132 | LEU 167  | GLU 251  |
| ASP 160 | MSE 170  | ASP 254  |
| ARG 166 | LEU 171  | LEU 255  |
| LEU 167 | LYS 174  |          |
| LEU 204 | PHE 208  |          |
| TRP 241 |          |          |
| ASN 244 |          |          |
| TRP 248 |          |          |

in sterol interaction are located on H6, H7, H8 and H10 and are all highly conserved in a sequence alignment of the closest BstC homologs. No hydrogen bond is anticipated between the sterol and protein in the docking models.

We then performed multiple 20 ns molecular dynamics simulations on the docked models of 4-monomethyl sterol from ICM-Pro. The simulations reveal that the substrate moves approximately 8–9 Å further into the binding site and the polar head makes a hydrogen bonding contact with the side chain of Glu86 (*Figure 5C*). This movement takes place almost immediately and the hydrogen bond persists for 94% of the total time. In both the docking models and MD simulations, the sterol is positioned near the entrance of CA1 from the C-terminal side. Next to the entrance is an extended loop that consists of highly conserved hydrophobic and neutral residues (residues 175–184).

Intriguingly, simulations with the 4,4-dimethyl sterol revealed that it behaves differently in that it moves slowly inwards and does not get as deep (~2.6 Å) into the pocket as the monomethyl sterol (*Figure 5 – figure supplement 5C*). A water-mediated contact is formed between the polar head and the side chain of Glu86 but persist for only 2% of the time. This lack of additional movement and H-bonding interaction may explain the binding data, where the protein has a $K_d$ for the monomethyl sterol that is ~30 × lower than for the dimethyl sterol substrate.

Mutagenesis to convert Glu86 to Gly and Trp to remove or reduce the potential interactions with the polar head of the sterols leads to ~4.5 and 100-fold decrease in the equilibrium affinity for 4-monomethyl sterol (*Figure 5—figure supplement 6A*). Meanwhile, the mutagenesis eliminates the binding to 4,4-dimethyl sterol, cholesterol and lanosterol (*Figure 5—figure supplement 6B and C*). These findings indicate that Glu86 plays a critical role in interacting with and stabilizing the sterols.

## Comparison to eukaryotic sterol transporters

In most eukaryotic cells, lipid transfer proteins (LTPs) are used to transfer sterols between subcellular membranes by non-vesicular transport to maintain sterol homeostasis in different sub-cellular organelles (*Baumann et al., 2005*). The most-studied family in mammals is the steroidogenic acute regulatory protein (StAR)-related lipid transfer (START) domain (STARD) proteins. All fifteen STARD proteins identified to date possess lipid-harboring START domains that can be divided into six subfamilies based on sequence similarity and ligand specificity. In addition to STARD proteins, a novel yet evolutionarily conserved family that contains the START-like domains called the GRAMD1s/Lam/Ltc family (GRAMD1s in mammals and Lam/Ltc in yeast) was shown to mediate non-vesicular sterol transport from the plasma membrane to the endoplasmic reticulum to maintain sterol homeostasis (*Gatta et al., 2015*). A third family consists of the Osh/ORP/OSBP proteins (Oxysterol-binding homology in yeast and Oxysterol-binding Protein/OSBP-Related Protein in mammals), which are considered to be either sterol-sensing and/or sterol-transfer proteins (*Raychaudhuri and Prinz, 2010*). Besides these, two Niemann-Pick type C proteins, NPC1 and NPC2, were also reported to directly bind and transport cholesterols and various oxysterols; defective NPC proteins causes NPC disease, a fetal neurodegenerative disorder where sterols aberrantly accumulate in the lysosome (*Carstea et al., 1997*).

To better understand the differences between eukaryotic and bacterial sterol transport, we compared the structures of BstB and BstC with well-studied eukaryotic sterol transporters from different families. Human STARD4 (PDB 6L1D), yeast Lam4 (PDB 6BYM), yeast Osh4 (PDB 1ZI7 and

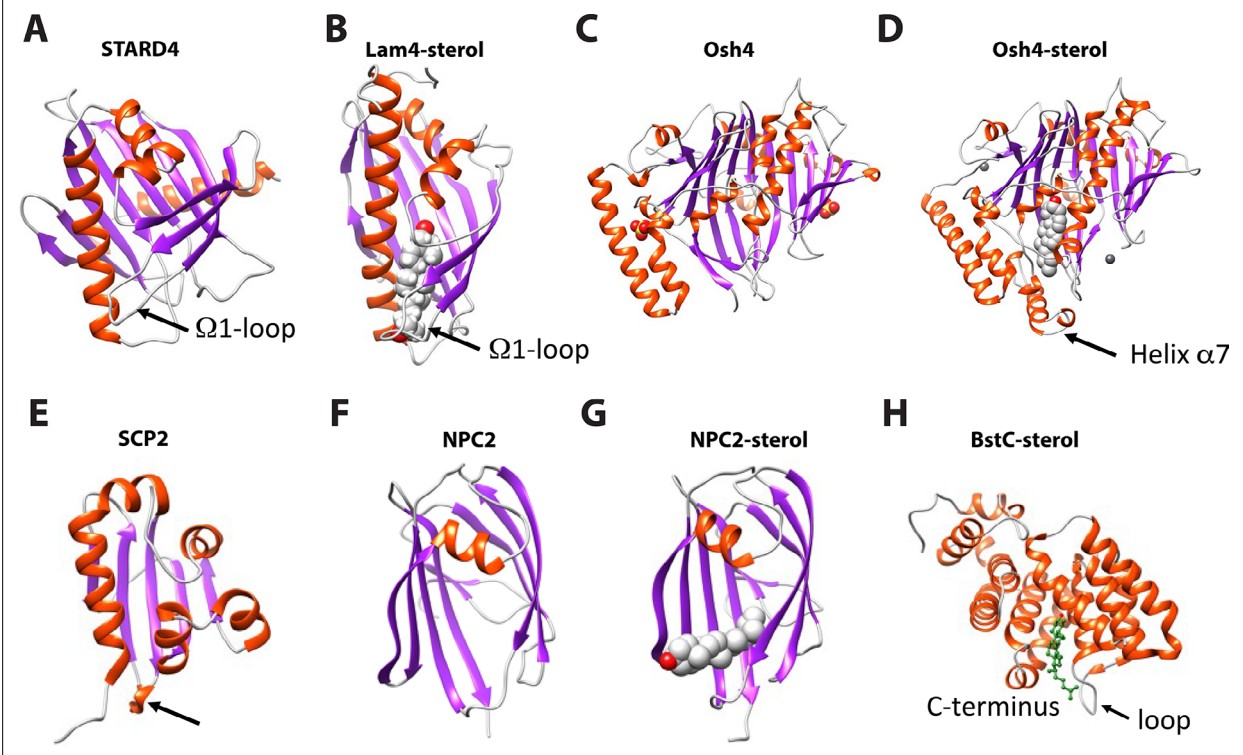

**Figure 6.** Structural comparison of eukaryotic sterol-binding proteins. (**A**) Apo human STARD4 (PDB 6L1D). (**B**) Yeast Lam4 bound to 25-hydroxycholesterol (PDB 6BYM). The Ω1-loop is depicted in structures that possess it. (**C–D**) Yeast apo- (**C**), PDB 1ZI7 and cholesterol-bound (**D**), PDB 1ZHY Osh4. The N-terminal helix α7 in Osh4, which is missing in the apo-Osh4 structure, may work similar to the Ω1-loop. (**E**) The non-specific LTP, rabbit SCP2 (PDB 1C44); (**F**) apo- (PDB 6R4M) and (**G**) ergosterol-bound (PDB 6R4N) NPC2. All structures are shown in cartoon representation with the α-helices and β-strands colored in red and magenta, respectively. The bound sterol substrates are depicted in grey spheres with the hydroxyl group(s) colored in red. (**H**) The Ω1-like loop in BstC docked complex with monomethyl sterol.

1ZHY), yeast NPC2 (PDB 6R4M and 6R4N), as well as a non-specific cytosolic sterol carrier protein, rabbit SCP2 (PDB 1C44) were chosen for the comparison. The crystal structures of STARD4, Lam4 and Osh4 show that they share a similar fold in that they are dominated by β-sheet motifs, which wrap around the longest α-helix and fold into a half-barrel/groove structure with the other side (or two) packed with several α-helices (*Figure 6A–D*). A central hydrophobic cavity can accommodate sterols. In most cases, the sterols are positioned proximal to the tunnel entrance with the hydroxyl end oriented towards the inside of the tunnel. Also, a 'lid' comprising a flexible 'Ω1' loop is observed near the entrance of tunnel in STARD4 and sterol-bound Lam4. This loop is replaced by an N-terminal α-helix in Osh4, which serves a similar function. The SCP2 structure also exhibits an α/β-fold with a 'lid'-like α-helix observed close to the N-terminus (*Figure 6E*). The sterol-bond NPC2 adopts an immunoglobulin-like β-sandwich fold with seven strands arranged into two β-sheets with a loosely packed hydrophobic cavity formed between the two sheets. The sterol is positioned at the entrance of the cavity with the hydroxyl end pointing out; no 'lid'-like structure is observed in NPC2 structure (*Figure 6F–G*).

Our comparative analysis shows that the eukaryotic sterol transporters share very little resemblance with BstB and BstC. The bilobed architecture of BstB is pronounced in its divergence from eukaryotic sterol transporters, suggesting that this protein evolved separately and may work in a distinct manner. For the all-helical BstC, the only similarities are the presence of a hydrophobic tunnel formed in the center of the protein that can accommodate sterols, and a loop next to the tunnel's entrance that may also be used as a 'lid' to restrict access to the substrate binding site (*Figure 6H*).

## Discussion

Since their discovery more than 40 years ago, the function of bacterial sterols has remained a mystery. Recent identification of bacterial sterol demethylase enzymes that diverge from their eukaryotic counterparts provided renewed enthusiasm for deciphering the molecular machinery involved in bacterial sterol synthesis and trafficking (*Lee et al., 2018*). The aerobic methanotroph *M. capsulatus* is known to synthesize 4-monomethyl sterol from 4,4-dimethyl sterol and to accumulate these sterols in its outer membrane. Until now, the transport process remained elusive owing to lack of knowledge of transporters that facilitate the trafficking. We now fill this gap with the identification of potential transporters of C-4 methyl sterols in *M. capsulatus*: BstA, BstB, and BstC.

Bioinformatic analyses revealed that BstA, BstB, and BstC belong to well-studied families of bacterial metabolite transporters. High resolution structures of BstB and BstC show that their large ligand binding sites display a dichotomy of hydrophobic and hydrophilic amino acids whose sidechains interact with the sterol hydrophobic tail and hydrophilic head. For BstC, this is especially interesting, as the only other family member characterized, Tp0956 from *Treponema pallidum*, displays a similar dichotomous ligand site. *Treponema pallidum*, the causative agent of syphilis, is a spirochete known to acquire and use host lipids– it is possible that Tp0956, and perhaps other T-component proteins, are involved in lipid trafficking.

In summary, the Bst proteins share sequence and structural homology with bacterial transporters involved in the trafficking of essential nutrients such as sugars, amino acids, vitamins, solutes, and metal ions (*Davidson et al., 2008*; *Higgins, 1992*). Their implication in the transport of sterols expands the substrate repertoire for their respective families. The substrate binding sites detected in BstB and BstC should now enable the curation of functionally homologous sites in bacterial proteins, including in those from human pathogens known to traffic host lipids.

## Materials and methods

### Bioinformatic analyses

The SSNs were generated using EFI-EST sever (https://efi.igb.illinois.edu/efi-est/). Proteins are represented by nodes. Only the nodes that share pairwise sequence alignments greater than the user-specified value will be connected by edges to create a network that reveals potential functional relationships among the proteins (*Atkinson et al., 2009*). HMMER webserver (https://www.ebi.ac.uk/Tools/hmmer/) was used to predict the superfamilies of Bst proteins. BstA is predicted to be a member of MMPL family (pfam03176 or IPR004869). Because some homologs are not annotated in the pfam family, we first used the Blast function to identify proteins sharing homology with BstA in UniProtKB (https://www.uniprot.org/). Homologs within the cutoff value of $5.3 \times 10^{-80}$ were also used in the SSN creation. The SSN was generated using the EFI-EST online tool (*Gerlt et al., 2015*) with an E-value threshold of $10^{-60}$ (~35% sequence identity); with this threshold, BstA proteins were grouped into a single cluster. UniRef50 (>50% sequence identity merged into a single node) was used to facilitate the calculation and save time during the SSN creation. BstB belongs to a family of ABC transporter phosphonate periplasmic substrate-binding proteins (pfam12974); this Pfam was selected to create an SSN at E-value threshold of $10^{-50}$ (~40% sequence identity). Homologous proteins found in the UniprotKB (cutoff at $1.6 \times 10^{-62}$) were added to the network as well. UniRef90 standard (≥90% sequence identity merged into a single node) was used to create the BstB SSN. BstC is classified as a member of the TRAP transporter T-component superfamily (IPR038537), which are thought to facilitate the import of small molecules into the cytoplasm (*Radolf et al., 2016*). An SSN was constructed with this dataset and the homologous genes found in UniprotKB (cut-off at $6.1 \times 10^{-39}$) at E-value threshold of $10^{-55}$ (~40% sequence identity). Cystoscape software (*Shannon et al., 2003*) was used to visualize the SSNs with the 'organic' layout. The genome neighborhood network and diagram were generated using the Enzyme Function Initiative-Genome Neighborhood Tool (EFI-GNT; https://efi.igb.illinois.edu/efi-gnt/) with the SSNs generated above. The neighborhood reading frame was set to 10 frames and the minimal co-occurrence was set to 20%.

### Attempted deletion of *M. capsulatus* Texas H156DRAFT_2759-2757

In attempts to generate the *M. capsulatus* Texas H156DRAFT_2759–2757 mutant, a linear DNA fragment was PCR-amplified from a plasmid containing a kanamycin resistance gene flanked by 1

kb *M. capsulatus* chromosomal sequence upstream of H156DRAFT_2759 and 1 kb downstream of H156DRAFT_2757. Electrocompetent *M. capsulatus* cells were prepared via the following protocol: a 300 mL culture was grown to mid-exponential phase, washed three times in 5% cold sterile glycerol and resuspended in 300 µL cold sterile 10% glycerol. The electrocompetent cells were mixed with 2.5 µL of the linear DNA fragment (1000 ng total) and shocked using various electroporation settings (Bacteria 1,2,3, and 5) on the Bio-Rad Pulser Xcell and the Ec1 setting on the Bio-Rad Micropulser [parameters were a combination of different cuvette gaps (0.1 cm or 0.2 cm), voltages (1.8–3.0 kV), resistances (200–400 Ω)]. Electroporated cells were grown up for 24 hr in 500 µL of media made with nitrate mineral salts (NMS) and methane; following this, the cells were plated on NMS-agar plates supplemented with kanamycin. No colonies were ever observed, suggesting that depletion of these transporters is toxic to the cells.

## Protein expression and purification

The genomic DNA of *M. capsulatus* Texas (ATCC 19069) was subjected to several rounds of PCR to isolate the genes encoding BstB and BstC without their respective signal peptides (cleavage sites determined by the SignalP 5.0 server; *Almagro Armenteros et al., 2019*). The final gene products were assembled into pET-28a vectors containing TEV-cleavable N-terminal hexa-histidine (His-) tags or pET-20b vectors containing C-terminal His-tags. DNA sequencing of the plasmids confirmed the correct insertion of the genes. The BstA$^{PD}$ gene was synthesized and inserted into pET-28a after codon optimization for expression in *E. coli*. The soluble periplasmic domain of BstA (BstA$^{PD}$) was designed based on the structure predicted by AlphaFold (*Jumper et al., 2021*). Briefly, two fragments spanning residues 54–277 and 510–732 were connected by a GSGSGSGS linker to replace the membrane-spanning sequence. The DNA encoding BstA$^{PD}$ was cloned into pET28a. All the protein sequences and primers are listed in *Table 6*.

BstA$^{PD}$, BstB (residues 24–281) and BstC (residues 16–264) proteins were expressed in *E. coli* BL21(DE3) cells and grown at 37 °C until OD$_{600nm}$ of 0.6. Protein expression was induced upon the addition of 0.25 mM isopropyl β-D-1thiogalactopyranoside (IPTG) while shaking at 18 °C for 12 hr. Cells were harvested through centrifugation at 8000×g for 15 min and resuspended in lysis buffer containing 25 mM Tris-HCl (pH 8.0), 100 mM NaCl, 2 mM 2-mercaptoethanol, and 1 mM phenyl-methylsulfonyl fluoride (PMSF). Cells were lysed by microfluidization and centrifuged at 11,000×g for 40 min to remove cell debris. The supernatant was applied to a Ni-NTA affinity column (HiTrap IMAC FF, Cytiva product # 17092104), washed with lysis buffer supplemented with 15 mM imidazole to remove weakly bound proteins, and eluted with lysis buffer containing 300 mM imidazole. The eluate was concentrated and applied to a size-exclusion column (HiLoad 16/600 Superdex 200 pg, Cytiva product # 28989335) to remove additional contaminants, and purity was assessed through gel electrophoresis.

Selenium-methionine (Se-Met) labeled BstB and BstC were produced by supplementing the endogenous methionine synthesis of the expression host with Se-Met. The cells were cultured in M9 minimal media until OD$_{600nm}$ of 0.4. L-lysine (100 mg/L), L-phenylalanine (100 mg/L), L-threonine (100 mg/L), L-isoleucine (50 mg/L), L-leucine (50 mg/L) and L-valine (50 mg/L) and L-Se-Met (60 mg/L) were added to the media. 0.25 mM IPTG was added after 1 hr to commence induction and cells were cultured for an additional 12 hr at 18 °C before harvesting. The proteins were purified by the protocol described above.

The N-terminal His-tags of proteins were removed during overnight incubation at room temperature with TEV protease at a molar ratio of 1:100 (TEV: protein). Proteins without tags were further purified by size-exclusion chromatography. The molecular weight of proteins was determined by SEC-MALS (Wyatt Technology). The purity of all purified proteins was assessed using SDS-PAGE stained with Coomassie Brilliant Blue, after which aliquots of the purest fractions were flash-cooled in liquid nitrogen and stored at –80 °C.

## Size-exclusion chromatography (SEC)-coupled multi-angle light scattering (MALS)

SEC-MALS was performed using an in-line Superdex 200 Increase 3.2/300 GL SEC column (GE Healthcare) combined with a miniDawn Multi-Angle Light Scattering (MALS) detector coupled with an Optilab refractive index detector (Wyatt Technology, Santa Barbara, CA, USA). A total of 15 µL (~200 ng)

protein was centrifuged at 13000 × $g$ for 10 min before being injected into the pre-equilibrated SEC column with buffer (25 mM Tris-HCl (pH 8.0), 100 mM NaCl and 2 mM 2-mercaptoethanol). Proteins were separated at a flow rate of 0.15 mL/min at room temperature. Molecular masses were calculated using the Astra6.1 software (Wyatt Technology).

## Sterols extraction from *M. capsulatus*

Lipids were extracted from cell pellets of 2 L *M. capsulatus* cultures using a modified Bligh-Dyer extraction method as previously described.(*Lee et al., 2018*) Cells were re-suspended in 10:5:4 (vol:vol:vol) methanol: dichloromethane (DCM):water and sonicated for 1 hr. The organic phase was separated with 1:1 (vol:vol) DCM:water, later washed three times with DCM and then transferred to a clean vial and dried with $N_2$ to yield the total lipids extraction (TLE). The TLE was further purified using Silica column chromatography to isolate the alcohol soluble lipids.(*Summons et al., 2013*) The TLE was added to a hand-packed silica column and lipids were eluted using solvent solutions of different polarities (Hexane, 8:2 Hexane:DCM, DCM, 1:1 DCM:Ethyl Acetate, Ethyl acetate). The alcohol soluble lipids (HS fraction) were eluted with 1:1 DCM:Ethyl Acetate. The HS fraction was then dried with $N_2$ and dissolved in DMSO. The 4-methylsterol and 4,4-dimethylsterol were purified from the alcohol soluble lipids by liquid-chromatography (LC) using an Agilent 1260 Infinity II LC System. Briefly, lipids were run through a InfinityLab Poroshell120 EC-C18 column (4.6x150 mm, 2.7 Micron). The solvent system consisted of Solvent A: MeOH (with.04% formic acid and 0.1% ammonia), Solvent B: acetonitrile, and Solvent C: Water (with.04% formic acid and 0.1% ammonia). The solvent gradient started with 84% Solvent A, 1% Solvent B, and 15% Solvent C and ended with 100% Solvent A over the course of 50 min. Isolated sterols were derivatized to trimethylsilyl ethers with 1:1 (vol:vol) Bis(trimethylsilyl)trifluoroacetamide: pyridine and confirmed using GC-MS analysis as described in *Lee et al., 2018*.

## Protein-lipid pull down assay

Recombinantly expressed and purified proteins (BstA[PD], BstB, BstC) were incubated with two different *M. capsulatus* lipid extracts: total lipid extractions (TLE) or the enriched sterol fraction of the TLE (HS), both dissolved in DMSO. The TLE (139.5 µg total) contained 65.7 ng of 4-monomethyl sterol and 338.9 ng of 4,4-dimethyl sterol. The HS fraction (99.5 µg total) contained 313.8 ng and 967 ng of 4-monomethyl sterol and 4,4-dimethyl sterol, respectively. The experimental reaction conditions are as follows: 40 µM protein (BstA[PD], BstB, BstC) was incubated with 7.5 µL of either the TLE or HS lipids. The reactions were carried out in reaction buffer of 200 mM HEPES pH 8.0, 100 mM NaCl and 0.05% Triton X-100. During the negative control reactions, the proteins were incubated with DMSO that did not contain any lipids. This serves as (1) a vehicle control since the *M. capsulatus* TLE and HS fractions were dissolved in DMSO and (2) to determine whether C4-methylsterols were bound to the Bst proteins as a result of protein expression in *E. coli*. The reactions (both control and experiment) were incubated at 25 °C for 20 hr. The proteins were purified from the reactions using HisPur Ni-NTA resin (Thermos Fisher), lipids from the eluted protein fractions were extracted using a modified Bligh Dyer protocol (*Ekiert et al., 2017*; *Lee et al., 2018*) and lipid samples were analyzed by GC-MS as previously described. HisPur Ni-NTA wash buffer: 25 mM Tris pH 8.0, 100 mM NaCl elution buffer: 25 mM Tris pH 8.0,100 mM NaCl, 300 mM imidazole. SDS PAGE electrophoresis was conducted to demonstrate that the fractions analyzed did in fact contain protein.

## GC-MS analysis of *M. capsulatus* lipids

Derivatized lipids were separated with an Agilent 7890B Series GC equipped with 2 Agilent DB-17HT columns in tandem (each 30 m x 0.25 mm i.d. x 0.15 µm film thickness) with helium as the carrier gas at a constant flow of 1.1 ml/min. The program was as follows: 100 °C for 2 min, then 12 °C/min to 250 °C and held for 10 min, then 10 °C/min to 330 °C and held for 17.5 min. Two µL of each sample was injected in splitless mode at 250 °C. The GC was coupled to an Agilent 5977 A Series MSD with the ion source at 230 °C and operated at 70 eV in EI mode scanning from 50 to 850 Da in 0.5 s. Lipids were identified based on their retention time and compared to previously published spectra (*Wei et al., 2016*).

## Microscale thermophoresis

Microscale thermophoresis (MST) experiments were performed with NanoTemper Monolith NT.115 instrument according to the manufacturer's instructions (NanoTemper Technologies). In brief, His-tagged BstA[PD], BstB and BstC were fluorescently labeled with Monolith His-Tag Labeling Kit RED-tris-NTA 2[nd] Generation according to the manufacturer's instructions. The 4-methylsterol and 4,4-dimethylsterol were extracted from *M. capsulatus*, while lanosterol (CAS:79630, Sigma-Aldrich) and cholesterol (CAS: AAA1147018, Fisher Scientific) were prepared in DMSO with a final stock concentration of 3.9 mM and 7.76 mM, respectively.

Fifty nM each of labeled proteins was diluted in assay buffer (20 mM HEPES pH 8.0, 100 mM NaCl, and 0.05% Tween 20), in the presence of 16-step serial dilution of 4-methylsterols or 4,4-dimethylsterols (ligand solubilized in DMSO at a final constant concentration of 5%). After incubation, samples were loaded into Monolith NT.115 Premium Capillaries (NanoTemper Technologies) and measurements were taken at a constant temperature of 23 °C. MST traces were collected with an LED excitation power of 40% and an MST laser power of 40%. The MO. Control Analysis software (NanoTemper) was used to analyze the interaction affinity and the equilibrium dissociation constant ($K_d$) for each ligand using the $K_d$ fit model or Hill fit model. The data was exported and plotted in Prism.

Equation of $K_d$ -model fitting:

$$f(c) = \frac{c + c_T + K_d - \sqrt{\left(c + c_T + K_d\right)^2 + 4\,c\,c_T}}{2\,c_T}$$

f(c) is the fraction bound at a given ligand concentration c. $K_d$ is the equilibrium dissociation constant and $c_T$ is the final concentration of target in the assay.

Equation for Hill-model fits:

$$f(c) = Unbound + \frac{Bound - Unbound}{1 + \left(\dfrac{EC_{50}}{C}\right)^{n_{Hill}}}$$

f(c) is the fraction bound at a given ligand concentration c. Unbound is the $F_{norm}$ signal of the target. Bound is the $F_{norm}$ signal of the complex. $EC_{50}$ is the half-maximal effective concentration and $n_{Hill}$ is the Hill coefficient. The Hill coefficient $n_{Hill}$ describes the degree of cooperativity of an interaction: $n_{Hill} > 1$ indicates positive cooperativity, while $n_{Hill} < 1$ indicates negative cooperativity.

## Crystallization of BstB and BstC

Sparse-matrix crystallization screens were performed with purified proteins at concentrations of about 5 mg/mL for each. All crystals were obtained by the sitting-drop vapor diffusion method by mixing 0.5 µL protein and 0.5 µL precipitant solution at 20 °C. Native and Se-met BstB crystals appeared in the MCSG1-C12 condition containing 25% (v/v) PEG3500 and 0.1 M Bis-tris pH 6.5 after 1 week of incubation at 20 °C. Crystals were cryoprotected in the same conditions with the addition of 20% (v/v) PEG400 before being flash-cooled and stored in liquid nitrogen. Native and Se-Met BstC crystals appeared in the MCSG2-B3 condition containing 20% (v/v) PEG3500 and 0.2 M ammonium citrate dibasic after 2 weeks of incubation at 20 °C. Crystals were cryoprotected in the same conditions with the addition of 33% (v/v) PEG400 before being flash-cooled and stored in liquid nitrogen. Dimer-BstC crystals are obtained in the MCSG2-A12 condition containing 40% (v/v) PEG 600 and 0.1 M Sodium Citrate: Citric Acid, pH 5.5 after 2 weeks of incubation at 20 °C. Crystals were flash-cooled and stored in liquid nitrogen.

## Data collection, processing, and structure determination

All the diffraction datasets were collected at 100 K at the Stanford Synchrotron Rdaiation Lightsource (SSRL, Stanford, CA, USA). BstB and dimer-BstC datasets were collected used a Pilatus 6 M detector at beamline BL9-2, while BstC datasets were collected at beamline BL14-1 with a MARCCD325 detector. Native datasets were collected at a fixed wavelength at 0.97946 Å. MAD datasets were collected on ligand-free SeMet-substituted crystals at three wavelengths near the Se K-edge (0.97896 Å, 0.91837 Å, and 0.97930 Å). All diffraction datasets were indexed and processed by XDS (*Kabsch, 2010a*; *Kabsch, 2010b*) and HKL3000 packages (*Minor et al., 2006*) The structures of BstB and BstC were phased by

the MAD method with Phenix AutoSol using the Se-anomalous diffraction (*Liebschner et al., 2019*). Subsequent density modification gave excellent electron-density maps, which allowed the building of the models. Iterations of refinement were carried out with Phenix-Refine, and model building was performed in Coot. The statistics of data collection and refinement are listed in *Table 3*.

All structural figures were prepared using PyMOL (*Schrödinger and DeLano, 2023*) or UCSF Chimera (*Pettersen et al., 2004*). Cavities in BstB were assessed using CASTp (*Tian et al., 2018*) with a probe radius of 1.4 Å, equivalent to the radius of water. The tunnel through BstC was identified with MOLE2.0 (*Pravda et al., 2018*) using default settings and a probe radius of 1.4 Å.

### Docking of substrate to BstB and BstC

Molecular docking studies were performed using ICM-Pro 3.8–6 a (*Abagyan et al., 1994b*; *Abagyan and Totrov, 1994a*). Briefly, chain A of the BstB and BstC crystal structure was used as the rigid receptor. Each protein molecule was converted to an ICM object, with optimization of hydrogen atom placement. Potential binding site were identified using the pocketfinder feature of ICM-Pro (*Abagyan and Kufareva, 2009*; *An et al., 2005*); for BstB and BstC the largest pockets had volumes of 1050 $A^3$ and 670 $A^3$, respectively. The coordinates of the C-4 methylated sterols (4-monomethylsterol and 4,4-dimethylsterol) were generated using the ICM-Pro ligand editing tools, based upon the structure of cholesterol. The ligands were docked into the binding pockets in both protein receptors. The MA-1–206 docking runs were performed multiple times, results were ranked in order to the overall score, and the most energetically favored binding modes were extracted from ICM-Pro as PDB files. Additional docking runs with BstB were carried out where three residues (Glu118, Tyr120 and Asn192) were allowed to have rotationally flexible side chains to simulate induced fit docking, using the explicit group docking feature of ICM-Pro, while the rest of the protein remained rigid. Result analyses and figure rendering were performed using PyMOL (Schrödinger and DeLano).

### Molecular dynamics simulations of BstB and BstC

Molecular dynamics (MD) simulations were performed in triplicate on models comprising BstB and BstC docked with 4-monomethylsterol and 4,4-dimethylsterol, using Desmond (*Bowers et al., 2006*) in the Schrodinger 2019–2 release. The docked complexes were prepared with Maestro (Schrodinger) using the OPLS3e force field (*Roos et al., 2019*). The pre-defined TIP3P water model (*Neria et al., 1996*) was used to build the system. The overall charge of the complexes was calculated as –1 and –10 for BstB and BstC respectively and neutralized with $Na^+$ ions. Prior to building in system for both complexes, 0.15 M salt (NaCl) was added, and the ions were restricted from coming within 20 Å of the ligand. The systems were minimized prior to the final 20 ns production step runs at 300 K and 1 atm pressure, using the Nosé–Hoover chain coupling scheme for the temperature control and the Martyna–Tuckerman–Klein chain coupling scheme with a coupling constant of 2.0 ps for pressure control (*Martyna et al., 1992*). Nonbonded forces were calculated using an r-RESPA integrator. The trajectories were saved at 10 ps intervals for analysis. For the 150 ns simulation on apo-BstB, the same protein preparation steps and variable settings were used except the trajectories were saved every 25 ps for analysis. Maestro and Desmond were run on the SHERLOCK 3.0 HPC cluster at Stanford University.

### Acknowledgements

This work was supported in part by NSF grant MCB 1919153 (to LMKD and PVW). LMKD was supported by funding from the Terman, Gabilan, and Hellman Fellowships, and is a MAC3 Impact Philanthropies Faculty Fellow. Crystallographic data were acquired at the Stanford Synchrotron Radiation Lightsource. Use of the Stanford Synchrotron Radiation Lightsource, SLAC National Accelerator Laboratory, is supported by the US Department of Energy, Office of Science, Office of Basic Energy Sciences under Contract No. DE-AC02-76SF00515. The SSRL Structural Molecular Biology Program is supported by the DOE Office of Biological and Environmental Research, and by the National Institutes of Health, National Institute of General Medical Sciences (P30GM133894). The contents of this publication are solely the responsibility of the authors and do not necessarily represent the official views of NIGMS or NIH.

## Additional information

### Competing interests
Laura MK Dassama: Reviewing editor, eLife. The other authors declare that no competing interests exist.

### Funding

| Funder | Grant reference number | Author |
|---|---|---|
| National Science Foundation | MCB 1919153 | Laura MK Dassama<br>Paula V Welander |

The funders had no role in study design, data collection and interpretation, or the decision to submit the work for publication.

### Author contributions
Liting Zhai, Amber C Bonds, Conceptualization, Resources, Data curation, Formal analysis, Validation, Investigation, Visualization, Methodology, Writing – original draft, Writing – review and editing; Clyde A Smith, Resources, Data curation, Formal analysis, Validation, Investigation, Visualization, Methodology, Writing – original draft, Writing – review and editing; Hannah Oo, Jonathan Chiu-Chun Chou, Resources, Data curation, Investigation; Paula V Welander, Conceptualization, Formal analysis, Supervision, Funding acquisition, Writing – original draft, Project administration, Writing – review and editing; Laura MK Dassama, Conceptualization, Data curation, Formal analysis, Supervision, Funding acquisition, Visualization, Methodology, Writing – original draft, Project administration, Writing – review and editing

### Author ORCIDs
Liting Zhai ⬛ http://orcid.org/0000-0003-3566-3472
Paula V Welander ⬛ http://orcid.org/0000-0002-9502-6902
Laura MK Dassama ⬛ https://orcid.org/0000-0002-0851-6373

Reviewer #1 (Public Review): https://doi.org/10.7554/eLife.90696.3.sa1
Reviewer #2 (Public Review): https://doi.org/10.7554/eLife.90696.3.sa2
Reviewer #3 (Public Review): https://doi.org/10.7554/eLife.90696.3.sa3
Author Response https://doi.org/10.7554/eLife.90696.3.sa4

## Additional files

### Supplementary files
• MDAR checklist

### Data availability
Crystallographic structures for BstB and BstC are deposited in the PDB (codes 7T1M and 7T1S).

The following datasets were generated:

| Author(s) | Year | Dataset title | Dataset URL | Database and Identifier |
|---|---|---|---|---|
| Zhai L, Dassama LMK | 2022 | Crystal structure of a bacterial sterol transporter | https://www.rcsb.org/structure/7T1M | RCSB Protein Data Bank, 7T1M |
| Zhai L, Dassama LMK | 2022 | Crystal structure of a bacterial sterol transporter | https://www.rcsb.org/structure/7T1S | RCSB Protein Data Bank, 7T1S |

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
