## [Editor Report · eLife assessment]

This is a **valuable** contribution to our understanding of how some bacteria can transport sterols from the cytoplasm to the outer membrane. Though much remains to be tested and explored, the data and analyses presented here provide **solid** evidence for the genetic and physical interaction of BstA/B/C with bacterially-produced sterols. The manuscript will be of interest to scientists focusing on the characterization of novel bacterial proteins and those studying lipid transport and acquisition in bacterial pathogens.

---

## [Referee Report · Reviewer #1 (Public Review)]

Summary

This article by Zhai et al, investigates sterol transport in bacteria. Synthesis of sterols is rare in bacteria but occurs in some, such as, M capsulatus where the sterols are found primarily in the outer membrane. In a previous paper the authors discovered an operon consisting of five genes, with two of these genes encoding demethylases involved in sterol demethylation. In this manuscript the authors set out to investigate the functions of the other three genes in the operon. Interestingly, through a bioinformatic analysis they show that they are an inner membrane transporter of the RND family, a periplasmic binding protein and an outer membrane associated protein, all potentially involved with lipid transport, so providing a means of transporting the lipids to the outer membrane. These proteins are then extensively investigated through lipid pulldowns, binding analysis on all three, and X-ray crystallography and docking of the latter two.

Strengths

The lipid pulldowns and associated MST binding analysis are convincing, clearly showing that sterols are able to bind to these proteins. The structures of BstB and BstC are high resolution with excellent maps that allow docking studies to be carried out. These structures are distinct from sterol binding proteins in eukaryotes.

Weaknesses

While the docking and molecular dynamics studies are consistent with the binding of sterols to BstB and BstC, this is not backed up particularly well. Their discussion, however, is measured and clearly provides a strong case for further investigation.

---

## [Referee Report · Reviewer #2 (Public Review)]

Summary:

In eukaryotes, sterols are crucial for signaling and regulating membrane fluidity, however, the mechanism governing cholesterol production and transport across the cell membrane in bacteria remains enigmatic. The manuscript by Zhai et al. sheds light on this topic by uncovering three potential cholesterol transport proteins. Through comprehensive bioinformatics analysis, the authors identified three genes bstA, bstB, and bstC encoding proteins which share homology with transporters, periplasmic binding proteins, and periplasmic components superfamily, respectively. Furthermore, the authors confirmed the specific interaction between these three proteins and C-4 methylated sterols and determined the structures of BstB and BstC. Combining these structural insights with molecular dynamics simulation, they postulated several plausible substrate binding sites within each protein.

Strengths:

The authors have identified 3 proteins that seem likely to be involved in sterol transport between the inner and outer membrane. The structures are of high quality, and the sterol binding experiments support a role for these proteins in sterol transport.

Weaknesses:

While the author's model is very plausible, direct evidence for a role of BstABC in transport, or that the 3 proteins function together in a single pathway, is limited.

---

## [Referee Report · Reviewer #3 (Public Review)]

Summary:

The work in this manuscript builds on prior efforts by this team to understand how sterols are biosynthesized and utilized in bacteria. The study reports a new function for three genes encoded near sterol biosynthesis enzymes, suggesting the resulting proteins function as a sterol transport system. Biochemical and structural characterization of the two soluble components of the pathway establishes that both proteins can bind sterols, with a preference for 4-methylated derivatives. High-resolution x-ray structures of the apoproteins reveal hydrophobic cavities of the appropriate size to accommodate these substrates. Docking and molecular dynamics simulations confirm this observation and provide specific insights into residues involved in substrate binding.

Strengths:

The manuscript is comprehensive and well-written. The annotation of a new function in a set of proteins related to bacterial sterol usage is exciting and likely to enable further study of this phenomenon - which is currently not well understood. The work also has implications for improving our understanding of lipid usage in general among bacterial organisms.

---

## [Author Response]

The following is the authors’ response to the original reviews.

**Public Reviews:**

**Reviewer #1 (Public Review):**
SummaryThis article by Zhai et al, investigates sterol transport in bacteria. Synthesis of sterols is rare in bacteria but occurs in some, such as M capsulatus where the sterols are found primarily in the outer membrane. In a previous paper the authors discovered an operon consisting of five genes, with two of these genes encoding demethylases involved in sterol demethylation. In this manuscript, the authors set out to investigate the functions of the other three genes in the operon. Interestingly, through a bioinformatic analysis, they show that they are an inner membrane transporter of the RND family, a periplasmic binding protein, and an outer membrane-associated protein, all potentially involved with lipid transport, so providing a means of transporting the lipids to the outer membrane. These proteins are then extensively investigated through lipid pulldowns, binding analysis on all three, and X-ray crystallography and docking of the latter two.StrengthsThe lipid pulldowns and associated MST binding analysis are convincing, clearly showing that sterols are able to bind to these proteins. The structures of BstB and BstC are high resolution with excellent maps that allow docking studies to be carried out. These structures are distinct from sterol-binding proteins in eukaryotes.

We thank the reviewer for their favorable impression of this work.

WeaknessesWhile the docking and molecular dynamics studies are consistent with the binding of sterols to BstB and BstC, this is not backed up particularly well. The MST results of mutants in the binding pocket of BstB have relatively little effect, and while I agree with the authors this may be because of the extensive hydrophobic interactions that the ligand makes with the protein, it is difficult to make any firm conclusions about binding.

We agree with the reviewer that at this point, there is no experimental evidence to define the sterol binding site in BstB. While in the manuscript we allude to the extensive hydrophobic interactions as being especially stabilizing and difficult to eliminate with one or two mutations, we are now also aware that hydrogen-bonding interactions with the polar head of the sterols are quite important (see data on BstC, where disruption of that interaction significantly reduces the equilibrium affinity for sterols). Our MD simulations show that at least 3 protein amino acids can participate in H-bonding with the sterols. Moreover, recent work from our lab show that even ligand site waters can extend an H-bonding network around the polar head of the lipid (Zhai et al., ChemBioChem 2023, 24, e202300156), thereby enabling H-bonding with amino acids that are further away from the ligand site. It is therefore difficult to predict which mutations will sufficiently destabilize the binding. While this question is one we will tackle in future studies focused on obtaining high-resolution substrate-bound structures of BstB or homologs, the findings reported here are still relevant and timely, and we posit will spur the discovery of functional homologs, including some in organisms that are more tractable.

The authors also discuss the possibility of a secondary binding site in BstB based on a slight cavity in domain B next to a flexible loop. This is not backed up in any way and seems unlikely.

The reviewer is correct in that the evidence for this second binding site weak. While the crystallographic structure shows a highly hydrophobic region and the binding studies suggests cooperativity exists in the binding of the 4methylsterol substrate, the docking studies do not strongly support binding at that site. As such, we have clarified in the manuscript that a second hydrophobic cavity is observed, but that its role in ligand interaction remains unexplored.

**Reviewer #2 (Public Review):**
Summary:In eukaryotes, sterols are crucial for signaling and regulating membrane fluidity, however, the mechanism governing cholesterol production and transport across the cell membrane in bacteria remains enigmatic. The manuscript by Zhai et al. sheds light on this topic by uncovering three potential cholesterol transport proteins. Through comprehensive bioinformatics analysis, the authors identified three genes bstA, bstB, and bstC encoding proteins which share homology with transporters, periplasmic binding proteins, and periplasmic components superfamily, respectively. Furthermore, the authors confirmed the specific interaction between these three proteins and C-4 methylated sterols and determined the structures of BstB and BstC. Combining these structural insights with molecular dynamics simulation, they postulated several plausible substrate binding sites within each protein.Strengths:The authors have identified 3 proteins that seem likely to be involved in sterol transport between the inner and outer membrane. The structures are of high quality, and the sterol binding experiments support a role for these proteins in sterol transport.

We thank the reviewer for this positive view of our work.

Weaknesses:While the author's model is very plausible, direct evidence for a role of BstABC in transport, or that the 3 proteins function together in a single pathway, is limited.

The reviewer is correct that we were unable to demonstrate that the three proteins work together to transport 4methylsterols. This is not for lack of trying. We first attempted gene deletion studies, and as mentioned in the manuscript (with more details now provided in the experimental section), this appeared to be lethal. We then attempted in vitro exchange experiments, in which the proteins would be used to transfer sterols from sterol-loaded “heavy” liposomes to a sterol-free “light” liposomes – such exchange assays are frequently performed with eukaryotic sterol transporters (see Chung et al., Science 2015, https://doi.org/10.1126/science.aab1370). These assays were not successful because (1) sterols incorporated poorly into liposomes made with *E. coli* polar lipids and yielded leaky liposomes; (2) use of liposomes prepared with the TLE of M. capsulatus proved more stable, but no appreciable exchange was observed; we reasoned that this might be due to the absence of an energy source for BstA, the RND component for which we have expressed and purified only the soluble periplasmic domain. Given the technical difficulty of these in vitro transport experiments, we will continue to pursue in vivo demonstration of function as new homologs are identified.

**Reviewer #3 (Public Review):**
Summary:The work in this manuscript builds on prior efforts by this team to understand how sterols are biosynthesized and utilized in bacteria. The study reports a new function for three genes encoded near sterol biosynthesis enzymes, suggesting the resulting proteins function as a sterol transport system. Biochemical and structural characterization of the two soluble components of the pathway establishes that both proteins can bind sterols, with a preference for 4methylated derivatives. High-resolution x-ray structures of the apoproteins reveal hydrophobic cavities of the appropriate size to accommodate these substrates. Docking and molecular dynamics simulations confirm this observation and provide specific insights into residues involved in substrate binding.Strengths:The manuscript is comprehensive and well-written. The annotation of a new function in a set of proteins related to bacterial sterol usage is exciting and likely to enable further study of this phenomenon - which is currently not well understood. The work also has implications for improving our understanding of lipid usage in general among bacterial organisms.

We thank the reviewer for this synopsis of our work.

Weaknesses:The authors might consider moving some of the bioinformatics figures to the main text, given how much space is devoted to this topic in the results section.

We have taken this advice and moved Figure S1 to the main manuscript.

**Reviewer #1 (Recommendations For The Authors):**
1. In the analysis of the MST data, the authors quote Hill coefficients. How reliable are these numbers? For BstB, for instance, it seems unlikely that more than one molecule would bind. Can the analysis be done without needing to include Hill coefficients?

We used fits that did and did not invoke cooperativity – see below. We are certain that both BstA and BstB are better fit with cooperativity invoked.

**Author response image 1. sa4fig1:** 

1. In looking at the maps associated with the structures, which were included in the review package, I see that two citric acid molecules fit beautifully into the density where currently PEG has been modelled. This needs to be fixed and some comments may be appropriate in the manuscript.

We thank the reviewer for calling our attention to this. Citric acid has now been added to the model, and we reason that these are present in the structure because citric acid was used in the crystallization condition. The revised model is now present in the PDB.

1. It is not necessary to show the two molecules in the asymmetric unit in Figure 4 given that it is not a dimer. This doesn't add anything to the manuscript.

We now show a single molecule of BstC in Figure 4 (now Figure 5).

1. I wouldn't consider the loops shown in Figure S4 as disordered. They have slightly higher B-values but are not completely mobile.

We did not refer to these loops as disordered. In the text, we say they “exhibit poor electron densities, suggesting conformational sampling of more than one state (Fig. S4A).”

**Reviewer #2 (Recommendations For The Authors):**
pg 7, "hinting at an astounding distinction": I might suggest a word other than astounding that conveys how statistically unlikely, unusual, etc. this result is.

Thank you – we have removed “astounding”.

pg 7, paragraph 2: Here the authors show that in the SSN analysis, BstB proteins cluster separately and suggest this implies a distinction in function. However, they also show that PhnD homologs do not cluster separately (distributed across multiple clusters), yet presumably have similar functions. I am not familiar with SSN, but it seems to me that the second statement about PhnD implies that the first statement about BstB might not be valid, i.e., if PhnD doesn't cluster based on function, on what basis can we conclude that BstB does? On what basis does clustering occur in the SSN analysis? Might it be driven by things other than function? This comment also concerns the final paragraph of this section.

The reviewer is correct in that PhnD homologs occupy separate clusters of the SSN. Many of these homologs were crystallized with phosphate-like compounds, but it is possible that they have non-overlapping substrate scopes and are therefore functionally distinct. As for the basis of clustering, the SSN is fully sequence-based. What has been observed is that proteins with highly similar sequences can have similar functions – but this is not always true.

pg 8, paragraph 1: The authors suggest that BstABC may be essential. This is probably not a critical claim and it might be simplest to just remove it, but if it is mentioned, the authors should probably explain what was attempted that failed, so a reader can assess the strength of the evidence supporting essentiality. For example, I don't see anything in the methods about genetic manipulations of M. capsulatus, so currently, this falls within the realm of "Data not shown".

We have provided additional information about the experimental techniques used to do this. This statement was included so that it is understood that the reason for the experimental failure is unlikely to be technical in nature, as we have successfully deleted some sterol related genes while others remain intractable.

Fig. 2A: It is unclear to me what is being plotted here, perhaps more experimental detail is required in the form of labels and/or legend. Is this a quantification of each sterol in each fraction separated by GC? There are essentially no methods provided for the GC-MS experiments. A reference is provided, but I think providing detailed methods for these specific experiments will provide a higher degree of scientific rigor. I am not sure what is standard for GCMS, but perhaps showing spectra in the supplement that establish the identity of the bound molecules as species I and II would be appropriate?

Additional experimental details have been provided and the figure legend changed to be more clear. Moreover, we now clearly state that the chromatograms shown were used to identify lipids due to retention times for spectra that were previously published in Wei et al., 2016.

pg 10-11, comparison with PhnD structure: Perhaps it is worth mentioning a 3rd possible explanation for the relative opening/closing of the cleft is simply crystal packing? I don't think it necessarily has to imply anything about a difference in function. Also, the focus seems to be on this pairwise comparison, but perhaps more insights could be gleaned from an analysis that included a wider range of homologs, especially if any are thought to bind hydrophobic substrates.

This could be true, and we have included a statement to that effect. We are unaware of homologs shown to bind to large, hydrophobic molecules.

I think that BstB is shown upside-down in sup movies relative to other figures. If it isn't changed, perhaps adding some labels would help orient the reader.

We have rotated the movies to be more consistent with the figures.

Fig. S7: No units are indicated for Kds (uM?).

Thank you – this has been fixed.

pg 11, paragraph 2. "adjacent to three residues: Glu118, Tyr120 and Asn192": The residue number used in the text doesn't seem to match the numbering in the PDB file. I think these residues correspond to Glu98, Tyr100, and Asn172 in the PDB file.

We regret this error. The correct numbering for both structures is now present in the deposited PDB files (7T1M for BstB and 7T1S for BstC).

pg 12, final paragraph: The authors present binding data for BstB variants with mutations in the putative sterol binding pocket identified in the structural and MD analyses. However, these mutants had no effect on binding. The authors rationalize this in terms of the size of the interface and hydrophobic nature (which indeed, may be correct and is very plausible), and it is worth noting that many of their mutations are to Ala and would largely preserve the hydrophobic nature of the cleft. However, these mutants raise questions about where sterols actually bind. No experimental evidence is presented that substrates bind in the cleft, it is only hypothesized based on structural homology, MD simulations, etc. These mutations formally provide evidence against the hypothesis being tested; I think that has to be discussed a bit more directly, alongside the caveats the authors already discuss about hydrophobicity, etc.

This is a valid point by the reviewer, and it is one we have attempted to address with our statement in the manuscript and in our response to reviewer 1. We have modified the relevant text to more clearly state that there is as of yet no experimental evidence for the binding of sterols to the cavity identified via molecular docking.

pg 13: Presumably this is not the full-length lipoprotein, but has been truncated/mutated in some way? Some statement of roughly what was purified/crystallized should be stated.

The SI methods on protein purification states that the genes of BstB and BstC without their respective signal peptides were obtained.

pg 13, last paragraph "TN1 exhibits hybrid hydrophobicity, with the sides horizontal to cavities being hydrophobic while the vertical sides are more hydrophilic". I don't really follow the horizontal vs vertical sides. Perhaps this could be described in a different way.

Noted and changed to “TN1 is closer to the N-terminal face of the structure, while CA1 and CA2 are proximal to the C-terminal face and form two open hydrophobic pockets; TN1 exhibits a mixture of hydrophobic and hydrophilic amino acids (Fig. 4B and Fig. S9B, Table S4).”

pg 15-16, "Comparison to eukaryotic sterol transporters": Perhaps this would be better suited for the discussion section? Could also be streamlined; it is mostly discussing and comparing eukaryotic sterol binding domains to each other, not to BstABC.

Given that BstB and BstC are the first identified proteins (and putative transporters) for bacterial sterol engagement, we thought a careful description of the existing sterol transporters (which are all eukaryotic) was warranted.

**Reviewer #3 (Recommendations For The Authors):**
I have just two minor suggestions for the authors if they wish to comment on or address them.1. Do the three proteins (BstA/B/C) form any sort of complex? Perhaps this property was not assessed - but it seemed possible that the B and C components might constitute a shuttle for the membrane-bound transporter?

This is an important observation – the unliganded version of these proteins show no appreciable affinity for each other. However, BstB (which would be expected to engage both with BstA and BstC) belongs to a family of proteins known to undergo significant conformational change upon substrate binding. It is possible that with substrate present, complexes are formed – we have yet to investigate this.

1. In Figure S1, panel C - it appears that the label for the BstC cluster may have migrated away from the intended location. In this figure, it might also be useful to indicate in the caption the meaning of the red coloring of the nodes?

The label is now fixed – thank you for drawing our attention to this.